# Linear Discriminant Analysis with High-dimensional Mixed Variables

Zhongqing Yang

School of Arts and Sciences

Guangzhou Maritime University, Guangzhou, China

Binyan Jiang

Department of Data Science and Artificial Intelligence

The Hong Kong Polytechnic University, Hong Kong, China

Chenlei Leng

Department of Applied Mathematics

The Hong Kong Polytechnic University, Hong Kong, China

Cheng Wang

School of Mathematical Sciences

Shanghai Jiao Tong University, Shanghai, China

Xinyang Yu *

Department of Statistics

London School of Economics and Political Science, London, United Kingdom

## Abstract

Datasets containing both categorical and continuous variables are frequently encountered in many areas. The dimensions of these variables can be very high especially in modern data analysis. Despite the recent progress made in modelling high-dimensional data for continuous variables, there is a scarcity of methods that can deal with a mixed set of variables. To fill this gap, this paper develops a novel approach for classifying high-dimensional observations with mixed variables. Our framework builds on a location model, in which the distributions of the continuous variables conditional on categorical ones are assumed Gaussian. We overcome the challenge of having to split data into exponentially many cells, or combinations of the categorical variables, by kernel smoothing, and provide new perspectives for its bandwidth choice to ensure an analogue of Bochner's Lemma, which is different to the usual bias-variance tradeoff. We show that the two sets of parameters in our model can be separately estimated and provide penalized likelihood for their estimation. Results on the estimation

*Corresponding author: X.Yu59@lse.ac.uk.

accuracy and the misclassification rates are established, and the competitive performance of the proposed classifier is illustrated by extensive simulation and real data studies.

**Keywords:** Bayes risk, High dimensional mixed variable, Linear discriminant analysis, Location model, Semiparametric estimation.

**Mathematics Subject Classification (2020):** 62H30

# 1 Introduction

Consider the problem of classifying very high-dimensional observations into categories. In a great many cases, the datasets can contain a mixed set of variables including discrete and continuous ones, both of which can be high-dimensional while the sample size is small. Examples include:

- In clinical practice, it is common to collect data that come with continuous variables and discrete variables. The dimension of these features can be high, while the number of patients is relatively small, especially for serious or rare diseases. For example, in the Hepatocellular Carcinoma dataset considered in the real data study, there are 165 patients with 22 continuous variables which are mainly from patients' medical test results, and 118 binary variables which are mainly indicators of related symptoms and medical histories.

- In integrative analysis, the main objective is to combine different datasets for a comprehensive study. One of the possible possibilities is to integrate continuous-type data with discrete-type data. For example, the Breast Cancer Gene Expression Profiles data considered in our analysis consists of 489 mRNA Z-Scores (which are measurements of the relative expression of patients' genes to the reference population), and a set of indicators of mutation for 173 genes. A strong motivation for combining these two datasets is that together they may provide more information about the mortality risk, the main quantity of interest.

To deal with these mixed variables, a simple strategy is to treat the categorical variables as continuous ones and apply existing classification methods developed for continuous variables. Such a treatment ignores the nature of categorical variables and intuitively incurs loss of information. Here we present a simple example. Consider a two-class classification problem where there are one continuous variable $X$ and one binary categorical variable $U$. Assume that the prior for the class label $L$ is balanced, i.e., $P(L=1) = P(L=2) = 0.5$, and that $P(U=0) = P(U=1) = 0.5$ for both classes. For Class 1, suppose the conditional distribution of $X$ given $U$ satisfies

$$X|U=0 \sim N(-1,1), \quad X|U=1 \sim N(1,1).$$

Likewise, for Class 2, assume that

$$X|U=0 \sim N(1,1), \quad X|U=1 \sim N(-1,1).$$

If we simply treat $U$ as a continuous variable that takes value 0 or 1, and seek for the best linear classifier, the misclassification rate for the optimal linear classifier will be easily seen as

$\frac{1}{2}\left[\frac{1}{2} + \Phi(-1)\right]$, which is more than twice of the optimal Bayes misclassification rate $\Phi(-1)$; see Section A.1 in the Supplementary for more details. An immediate message from this simple example is that, in order to obtain a sound classifier, we may have to handle the effects of categorical variables and continuous variable differently, and seek for ways of capturing their interactions. In our setting, the challenge on the need to handle mixed variables is further exasperated by the high-dimensionality of the problem.

**The setup**. Denote $(\mathbf{Z}, \mathbf{U})$ as the mixed variable, where $\mathbf{Z}$ is a $p$-dimensional continuous variable and $\mathbf{U}$ is a $d$-dimensional discrete variable. Both $p$ and $d$ are very large. For simplicity, we shall assume that all variables in $\mathbf{U}$ are binary. Note that variables with more than two categories can be transformed to be binary by introducing a set of dummy (binary) variables (Krzanowski, 1993). Write the class label as $L \in \{1, 2\}$ and denote the probability function of $(\mathbf{Z}, \mathbf{U})$ in class $i$ as $f_i(\mathbf{Z}, \mathbf{U})$ for $i = 1, 2$. The Bayes rule, which is the optimal rule achieving the smallest misclassification rate among all discriminant rules, classifies a data point to the first class if and only if

$$\pi_1 f_1(\mathbf{Z}, \mathbf{U}) > \pi_2 f_2(\mathbf{Z}, \mathbf{U}),$$

where $P(L = i) = \pi_i, i = 1, 2$, is the prior probability of an observation coming from class $i$. As an application of the Bayes rule, consider the case where there are no discrete variables. If we assume that observations from class $i$ follow $N(\mu_i, \Sigma)$, a multivariate Gaussian distribution with class-specific mean $\mu_i \in \mathbb{R}^p$, and a common covariance matrix $\Sigma \in \mathbb{R}^{p \times p}$, an application of the Bayes rule gives rise to the familiar linear discriminant analysis (LDA) rule which assigns observation to class 1 if and only if

$$(\mu_1 - \mu_2)^\top \Sigma^{-1} \left[\mathbf{Z} - \frac{\mu_1 + \mu_2}{2}\right] + \log \frac{\pi_1}{\pi_2} > 0. \tag{1}$$

The Gaussianity assumption of the observations with a common covariance matrix is particularly appealing since the resulting Bayes rule is a simple linear function of the variable with an index $\Sigma^{-1}(\mu_1 - \mu_2)$ and an intercept $\log\{\pi_1/\pi_2\}$. Indeed, a widely studied and popular approach in high-dimensional classification, as discussed in previous work below, is to assume a sparse index that gives rise to the so-called sparse LDA (Cai and Liu, 2011; Fan et al., 2012; Mai et al., 2012; Mai and Zou, 2013).

**A semiparametric location model**. The presence of the discrete variables $\mathbf{U}$ clearly rules out the use of a joint Gaussian distribution for $(\mathbf{Z}, \mathbf{U})$. To continue to enjoy the attractive properties of the Gaussian distribution for classification, we employ an idea originated in Olkin and Tate (1961) by assuming a location model. Specifically, in this model, we treat the discrete random vector $\mathbf{U}$ as a location or cell, and assume that conditional on $\mathbf{U}$, the continuous random vector $\mathbf{Z}$ follows a location-dependent multivariate normal distribution in that $\mathbf{Z}|\mathbf{U} \sim N(\mu_i(\mathbf{U}), \Sigma(\mathbf{U})), i = 1, 2$. The common discrete variable dependent covariance $\Sigma(\mathbf{U})$ is inspired by the assumption in LDA. Denote the probability that an observation from class $i$ falls in cell $\mathbf{U} = \mathbf{u}$ as $p_i(\mathbf{u})$ and the posterior probability of $(\mathbf{Z}, \mathbf{U})$ from class $i$ as $P(i|\mathbf{Z}, \mathbf{U}), i = 1, 2$. Under

this location model, the optimal Bayes rule classifies $(\mathbf{Z}, \mathbf{U})$ into class 1 if

$$\log \frac{P(1|\mathbf{Z}, \mathbf{U})}{P(2|\mathbf{Z}, \mathbf{U})} = \log \frac{\pi_1 f_1(\mathbf{Z}, \mathbf{U})}{\pi_2 f_2(\mathbf{Z}, \mathbf{U})} \tag{2}$$
$$= [\mu_1(\mathbf{U}) - \mu_2(\mathbf{U})]^\top \Sigma^{-1}(\mathbf{U}) \left[ \mathbf{Z} - \frac{\mu_1(\mathbf{U}) + \mu_2(\mathbf{U})}{2} \right] + \log \frac{\pi_1 p_1(\mathbf{U})}{\pi_2 p_2(\mathbf{U})} > 0,$$

and into class 2 otherwise. Denote $\beta(\mathbf{U}) := \Sigma^{-1}(\mathbf{U})[\mu_1(\mathbf{U}) - \mu_2(\mathbf{U})]$, and $\eta(\mathbf{U}) := \log \frac{\pi_1 p_1(\mathbf{U})}{\pi_2 p_2(\mathbf{U})}$. The classifier in (2) can be written as

$$\beta(\mathbf{U})^\top \left[ \mathbf{Z} - \frac{\mu_1(\mathbf{U}) + \mu_2(\mathbf{U})}{2} \right] + \eta(\mathbf{U}) > 0, \tag{3}$$

which is a functional linear classifier in $\mathbf{Z}$, with a location-dependent direction $\beta(\mathbf{U})$ and a location-dependent intercept $\eta(\mathbf{U})$. In our setting, $\beta(\mathbf{u})$ is a $p$-dimensional vector for each fixed $\mathbf{u}$, and $\mathbf{u}$ itself is also high-dimensional. As a result, the dimensionality of the model is extremely high. In the setting considered in this paper where $\mathbf{U}$ is a vector of binary variables,

- For $\beta(\mathbf{U})$, we have $2^d$ $p$-dimensional vectors to estimate;

- For $\eta(\mathbf{U})$, we have $2^d$ different values to estimate.

Thus, even after assuming the location model, our estimation problem is much more challenging than that of the LDA in which only a scalar intercept and a $p$-dimensional vector need estimating. Given a relative small sample size, there is no hope that either $\beta(\mathbf{U})$ or $\eta(\mathbf{U})$ can be reasonably estimated unless some kind of structures are assumed. In this paper, we focus on a general scenario where $\beta(\mathbf{U})$ is treated as a function which varies smoothly over the location $\mathbf{U}$, and $\eta(\mathbf{U})$ is modelled by a parametric first order approximation. Some of the properties regarding the location model (e.g., Proposition 2.1) and our theoretical framework can be adapted to cases where more stringent assumptions are imposed to further simplify the model complexity; see Section 5 for discussion. Because of the dependence of $\beta(\mathbf{U})$ on $U$, we shall refer to our model as the semiparametric location model.

**Estimation**. Denote the sample from population 1 by $(\mathbf{U}_1, \mathbf{X}_1), \ldots, (\mathbf{U}_{n_1}, \mathbf{X}_{n_1})$ and the sample from population 2 by $(\mathbf{V}_1, \mathbf{Y}_1), \ldots, (\mathbf{V}_{n_2}, \mathbf{Y}_{n_2})$, and write $n = n_1 + n_2$. We briefly summarize the estimation strategy here, leaving the detailed construction to Section 2.2. A key observation, formalized in Proposition 2.1, is that the classification direction $\beta(\mathbf{U})$ and the intercept $\eta(\mathbf{U})$ can be estimated separately.

For the intercept, Proposition 2.1 gives

$$\eta(\mathbf{u}) = \log \frac{P(L = 1 \mid \mathbf{U} = \mathbf{u})}{P(L = 2 \mid \mathbf{U} = \mathbf{u})},$$

so that $\eta(\mathbf{u})$ is the logit transformation of the conditional class probability. Since $\eta(\mathbf{u})$ is a function on $\{0,1\}^d$, it can be represented as a polynomial in $\mathbf{u} = (u_1, \ldots, u_d)$. In the high-dimensional setting considered here, we adopt a parsimonious first-order approximation

$$\eta(\mathbf{u}) = A_0 + \sum_{i=1}^{d} A_i u_i. \tag{4}$$

This form is also natural for binary variables: (4) holds, for example, when the higher-order coefficients in the polynomial expansions of $\log\{\pi_1 p_1(\mathbf{u})\}$ and $\log\{\pi_2 p_2(\mathbf{u})\}$ coincide. Further discussion of (4) is given in Supplementary Section A.2. Under (4), we estimate $(A_0, \mathbf{A})$, where $\mathbf{A} = (A_1, \ldots, A_d)^\top$, by the penalized logistic loss

$$
\begin{aligned}
(\hat{A}_0, \hat{\mathbf{A}}) \quad = \quad & \underset{A_0 \in \mathbb{R}, \, \mathbf{A} \in \mathbb{R}^d}{\arg\min} \frac{1}{n} \bigg\{ \sum_{i=1}^{n_1} \Big[ -(A_0 + \mathbf{A}^\top \mathbf{U}_i) + \log\{1 + \exp(A_0 + \mathbf{A}^\top \mathbf{U}_i)\} \Big] \\
& + \sum_{j=1}^{n_2} \log\{1 + \exp(A_0 + \mathbf{A}^\top \mathbf{V}_j)\} \bigg\} + \lambda_\eta |\mathbf{A}|_1,
\end{aligned}
\tag{5}
$$

where $|\mathbf{A}|_1$ denotes the $\ell_1$ norm and $\lambda_\eta > 0$ is a tuning parameter.

We next estimate the location-dependent classification direction

$$
\beta(\mathbf{U}) = \Sigma^{-1}(\mathbf{U})\{\mu_1(\mathbf{U}) - \mu_2(\mathbf{U})\}.
$$

Since $\mu_1(\mathbf{u})$, $\mu_2(\mathbf{u})$ and $\Sigma(\mathbf{u})$ vary with the high-dimensional discrete location $\mathbf{u}$, we first estimate them by kernel smoothing over the Hamming cube. Denote the resulting estimators by $\hat{\mu}_1(\mathbf{u})$, $\hat{\mu}_2(\mathbf{u})$ and $\hat{\Sigma}(\mathbf{u})$. For a fixed $\mathbf{u}$, $\beta(\mathbf{u})$ is the minimizer of the convex quadratic loss

$$
b^\top \Sigma(\mathbf{u}) b - 2 b^\top \{\mu_1(\mathbf{u}) - \mu_2(\mathbf{u})\}.
$$

We therefore estimate $\beta(\mathbf{u})$ by

$$
\hat{\beta}(\mathbf{u}) := \underset{b \in \mathbb{R}^p}{\arg\min} \left[ b^T \hat{\Sigma}(\mathbf{u}) b - 2 b^T \{\hat{\mu}_1(\mathbf{u}) - \hat{\mu}_2(\mathbf{u})\} + \lambda_\beta |b|_1 \right],
\tag{6}
$$

where $\lambda_\beta > 0$ is a tuning parameter. The kernel estimators $\hat{\mu}_1$, $\hat{\mu}_2$ and $\hat{\Sigma}$, as well as the bandwidth selection, are defined in Section 2.2. Because $\beta(\mathbf{U})$ and $\eta(\mathbf{U})$ can be estimated separately, the tuning parameters $\lambda_\beta$ and $\lambda_\eta$ can also be selected separately. In particular, Proposition 2.1 implies that $\lambda_\beta$ can be chosen by minimizing the classification error with the intercept temporarily set to zero.

**Our contributions.** Our methodological contribution is to propose a semiparametric location model for classifying high-dimensional mixed variables in a unified framework. The proposed classifier preserves the LDA-type structure for the continuous variables conditional on the discrete location, while allowing the classification direction and intercept to depend on that location. We show that the classification direction $\beta(u)$ and the intercept $\eta(u)$ can be estimated separately, which leads to a tractable estimation procedure. On the theoretical side, we establish concentration inequalities for the kernel estimators over the high-dimensional Hamming cube and derive estimation error bounds and the asymptotic misclassification rate of the proposed classifier.

**Prior work.** High-dimensional datasets containing both discrete and continuous variables are frequently encountered in the big data era. For discriminant analysis, while there are early works in investigating how to model mixed variables under the traditional fixed dimensional setting,

recent research has focused almost exclusively on datasets with high dimensional continuous variables. Methodologies that can effectively take into account both the high-dimensionality and the mixing nature of the datasets are scarce.

When the dimensionality is fixed, discriminant analysis with discrete and continuous data has been well studied. In addition to the simple strategy by transforming the two different types of variables into one type, as mentioned previously, another approach is to establish different discriminant rules for different types of variables and combine them to obtain a final classifier; see for examples Cochran and Hopkins (1961) and Xu et al. (1992). Krzanowski (1975, 1980) first proposed linear discriminant rules based on the location model (2). Later, more comprehensive discriminant rules were proposed based on logistic discrimination, kernel estimation and the location model; see for example Aitchison and Aitken (1976), Knoke (1982), Asparoukhov and Krzanowski (2000), Kokonendji and Ibrahim (2016) and the references therein. Variable selection for location model based discriminant rules and further extension to quadratic rules and a nonparametric smoothing version can be found in Daudin (1986), Krzanowski (1993), Krzanowski (1994), Krzanowski (1995), Asparoukhov and Krzanowski (2000) and Mahat et al. (2007). Clearly, these approaches developed under fixed dimensionality are not applicable any more to the modern data era where dimensions of the data are high. In particular, in terms of theoretical analysis, these discriminant rules are either algorithmic without theoretical justification, or statistically justified under the fixed dimensional assumption.

When the dimensionality is high with discrete variables absent, there is a large growing literature devoted to the study of classification. Bickel and Levina (2004) first showed that using estimates developed under the fixed-dimensionality scenario for high-dimensional problems gives a classifier equivalent to random guessing in the worst case scenario. For the LDA in (1), there are many approaches proposed to deal with the high-dimensionality. Under suitable sparsity conditions on $\mu_1$, $\mu_2$ and $\Sigma$, Shao et al. (2011) proposed to shrink the entries of their empirical estimates. By assuming that $\Sigma^{-1}(\mu_1 - \mu_2)$ is sparse, several papers proposed to estimate this quantity by minimizing a penalized loss function that constrains its $\ell_1$ norm (Cai and Liu, 2011; Fan et al., 2012; Mai et al., 2012; Mai and Zou, 2013). Mai et al. (2019) further studied a multiclass extension of the LDA. Jiang et al. (2018) proposed a sparse quadratic discriminant analysis method that allows the within-class covariance matrices to differ. Jiang et al. (2020) investigated a scenario where the covariance matrices are varying with a fixed-dimensional continuous variable.

Despite these new developments in high dimensional LDA for model (1), it is challenging to develop proper estimation procedures for (2), when the dimensions of the continuous and discrete variables, namely $p$ and $d$, are both large. On one hand, for a given $d$, there will be $2^d$ locations and hence there will be a lot of empty cells unless the sample size grows exponentially in $d$ (Hall, 1981). On the other hand, the means and covariance matrix in (2) are now functions of the location, i.e., the high dimensional discrete variable. Of all the papers reviewed, Jiang et al. (2020) is the closest to this paper. However, unlike the dynamic LDA in that paper where the index variable is defined on a compact, continuous and finite dimensional space, the space $\{0,1\}^d$ is irregular and as $d$ tends to infinity, one would encounter a small ball probability issue analogous to that occurs in infinite dimensional spaces (Hong and Linton, 2016), bringing an

extra layer of complexity in establishing theoretical properties.

**Content of the paper**. The remainder of this paper is organized as follows. In Section 2, we introduce the separability property of the semiparametric location model, and provide more details on the estimation of its parameters. In Section 3, we provide consistency results for the estimation of $\beta(\mathbf{u})$ and $\eta(\mathbf{u})$, and evaluate the asymptotic misclassification rate of the estimated classifier. In Section 4, we conduct extensive numeric studies on simulated data and seven real datasets to illustrate the competitive performance of our method, with comparison to some modern approaches. Concluding remarks and further discussions are provided in Section 5. Technical proofs are deferred to the Supplementary.

## 2 Semiparametric Location Model: Estimation

In our semiparametric location classifier in (3), $\eta(\mathbf{u})$ is a linear function of $\mathbf{u}$ and $\beta(\mathbf{u})$ is a function of the location $\mathbf{u}$. In this section we will first look at the classification problem from the perspective of minimizing the expected misclassification rate, from which it is found that although $\beta(\mathbf{u})$ and $\eta(\mathbf{u})$ both appear in the Bayes rule (3), they can be independently estimated. We then focus on discussing the estimation of $\beta(\mathbf{u})$, as the problem of estimating $\eta(\mathbf{u})$ has been discussed in the Introduction.

### 2.1 Separability of $\beta(\mathbf{u})$ and $\eta(\mathbf{u})$

For a given location $\mathbf{U} = \mathbf{u}$, consider a general discriminant rule

$$D(\mathbf{b}(\mathbf{u}), b_0(\mathbf{u})) := \mathbf{b}(\mathbf{u})^\top [\mathbf{Z} - \frac{\mu_1(\mathbf{u}) + \mu_2(\mathbf{u})}{2}] + b_0(\mathbf{u}), \tag{7}$$

which classifies a new observation $(\mathbf{Z}, \mathbf{u})$ to class 1 if and only if $D(\mathbf{b}(\mathbf{u}), b_0(\mathbf{u})) > 0$. Here (7) is a general discriminant rule with arbitrary functions $b(u)$ and $b_0(u)$. It should be distinguished from (3), which is the Bayes classifier under the location model and is written in terms of the true quantities $\beta(u)$ and $\eta(u)$. Under the location model, the misclassification rate of the classifier $D(\mathbf{b}(\mathbf{u}), b_0(\mathbf{u}))$ over all locations can be seen as

$$R(D(\mathbf{b}(\mathbf{u}), b_0(\mathbf{u})) = \sum_{\mathbf{u} \in \{0,1\}^d} [\pi_1 p_1(\mathbf{u}) P_{\mathbf{u}}(2|1) + \pi_2 p_2(\mathbf{u}) P_{\mathbf{u}}(1|2)], \tag{8}$$

where $P_{\mathbf{u}}(i|j)$ is the conditional misclassification probability of classifying $\mathbf{Z}$ from class $i$ to class $j$ given the location $\mathbf{u}$. Let $\Phi(\cdot)$ be the cumulative distribution function of the standard normal distribution. Under the Gaussianity assumption we have

$$P_{\mathbf{u}}(i|j) = \Phi \left( \frac{b_0(\mathbf{u}) - \mathbf{b}(\mathbf{u})^\top [\mu_i(\mathbf{u}) - \mu_j(\mathbf{u})]/2}{\sqrt{\mathbf{b}(\mathbf{u})^\top \Sigma(\mathbf{u}) \mathbf{b}(\mathbf{u})}} \right). \tag{9}$$

The purpose of classification is to seek a classification direction $\mathbf{b}(\mathbf{u})$ and the corresponding intercept $b_0(\mathbf{u})$ such that the expected misclassification rate in (8) is minimized. Although the estimation of these two arguments are interrelated, we show in the following proposition that

their estimation can be separated.

**Proposition 2.1.** *Assume the location model hold, and let $\beta(\mathbf{u})$ and $\eta(\mathbf{u})$ be the optimal classification direction and intercept in the Bayes classifier* (3), *respectively. Consider $D(b(\mathbf{U}), 0)$, a special case of the general discriminant rule* (7) *with a zero intercept: $b_0(\mathbf{u}) = 0$. We have*

$$\beta(\mathbf{u}) = \Sigma^{-1}(\mathbf{u})[\mu_1(\mathbf{u}) - \mu_2(\mathbf{u})] = \arg\min_{\mathbf{b}(\mathbf{u}) \in \mathcal{D}} E\{R(\mathbf{b}(\mathbf{u}), 0)\},$$

*where $\mathcal{D}$ is the set of all functions from $\{0, 1\}^d$ to $\mathbb{R}$, and $E$ is the expectation. On the other hand, for the optimal intercept $\eta(\mathbf{u})$ we have: $\eta(\mathbf{u}) = \log \frac{P(L=1|\mathbf{U}=\mathbf{u})}{P(L=2|\mathbf{U}=\mathbf{u})}$.*

This proposition ensures that the estimation $\beta(\mathbf{u})$ and $\eta(u)$ can be conducted separately. In particular, it indicates that the estimation of $\beta(\mathbf{u})$ can be conducted by simply setting $\eta(\mathbf{u}) = 0$.

## 2.2 Estimation of $\beta(\mathbf{u})$

To estimate $\beta(\mathbf{u})$ at any cell $\mathbf{u}$, we will resort to kernel smoothing. Recall that our sample consists of $(\mathbf{U}_1, \mathbf{X}_1), \ldots, (\mathbf{U}_{n_1}, \mathbf{X}_{n_1})$ from population 1 and $(\mathbf{V}_1, \mathbf{Y}_1), \ldots, (\mathbf{V}_{n_2}, \mathbf{Y}_{n_2})$ from population 2. For any $\mathbf{u}_1, \mathbf{u}_2 \in \{0, 1\}^d$, define the normalized Hamming distance between $\mathbf{u}_1$ and $\mathbf{u}_2$ as $< \mathbf{u}_1, \mathbf{u}_2 >= \frac{|\mathbf{u}_1 - \mathbf{u}_2|_1}{d}$, where $|\cdot|_1$ is the $\ell_1$ norm. Our estimation of the mean and covariance matrix is based on the following Nadaraya-Watson type local smoothing. For given bandwidths $h_x$ and $h_y$, we estimate $\mu_1$ and $\mu_2$ as

$$\hat{\mu}_1(\mathbf{u}) = \sum_{j=1}^{n_1} \frac{\exp\{-(dh_x)^{-1}|\mathbf{U}_j - \mathbf{u}|_1\}\mathbf{X}_j}{\sum_{j=1}^{n_1} \exp\{-(dh_x)^{-1}|\mathbf{U}_j - \mathbf{u}|_1\}}, \quad \hat{\mu}_2(\mathbf{u}) = \sum_{j=1}^{n_2} \frac{\exp\{-(dh_y)^{-1}|\mathbf{V}_j - \mathbf{u}|_1\}\mathbf{Y}_j}{\sum_{j=1}^{n_2} \exp\{-(dh_y)^{-1}|\mathbf{V}_j - \mathbf{u}|_1\}}.$$

Based on these, we shall establish the theory for the kernel smoothing estimators. Note that $\Sigma(\mathbf{u}) = E[\mathbf{X}(\mathbf{u})\mathbf{X}(\mathbf{u})^\top] - E\mathbf{X}(\mathbf{u})(E\mathbf{X}(\mathbf{u}))^\top = E[\mathbf{Y}(\mathbf{u})\mathbf{Y}(\mathbf{u})^\top] - E\mathbf{Y}(\mathbf{u})(E\mathbf{Y}(\mathbf{u}))^\top$. We estimate $\Sigma(\mathbf{u})$ as $\hat{\Sigma}(\mathbf{u}) = \frac{n_1}{n}\hat{\Sigma}_1(\mathbf{u}) + \frac{n_2}{n}\hat{\Sigma}_2(\mathbf{u})$, where

$$\hat{\Sigma}_1(\mathbf{u}) = \frac{\sum_{j=1}^{n_1} \exp\{-(dh_{xx})^{-1}|\mathbf{U}_j - \mathbf{u}|_1\}\mathbf{X}_j\mathbf{X}_j^T}{\sum_{j=1}^{n_1} \exp\{-(dh_{xx})^{-1}|\mathbf{U}_j - \mathbf{u}|_1\}} - \hat{\mu}_1(\mathbf{u})\hat{\mu}_1(\mathbf{u})^\top,$$

$$\hat{\Sigma}_2(\mathbf{u}) = \frac{\sum_{j=1}^{n_2} \exp\{-(dh_{yy})^{-1}|\mathbf{V}_j - \mathbf{u}|_1\}\mathbf{Y}_j\mathbf{Y}_j^T}{\sum_{j=1}^{n_2} \exp\{-(dh_{yy})^{-1}|\mathbf{V}_j - \mathbf{u}|_1\}} - \hat{\mu}_2(\mathbf{u})\hat{\mu}_2(\mathbf{u})^\top,$$

with the bandwidth parameters $h_{xx}$ and $h_{yy}$ controlling the smoothness of the estimators. Then $\beta(\mathbf{u})$ is estimated via the minimization problem in (6).

To make the implementation of the proposed method more explicit, we summarize the main computational steps in Algorithm 1.

## 3 Theoretical Results

The following notations will be used. For a $k \times p$ matrix $\mathbf{M} = (M_{ij})_{k \times p}$ we denote the vector $l_\infty$ norm induced matrix norm as $\|M\|_L := \max_{1 \leq i \leq k} \sum_{j=1}^p |M_{ij}|$, and denote $\|\mathbf{M}\|_\infty = \max_{1 \leq i \leq k, 1 \leq j \leq p} |M_{ij}|$. Write the $j$th component of $\mathbf{X}_i$ as $X_{ij}$ and define $Y_{ij}$ likewise. With

**Algorithm 1** Semiparametric location classifier

**Input** Training data $\{(U_i, X_i), L_i = 1\}_{i=1}^{n_1}$ and $\{(V_j, Y_j), L_j = 2\}_{j=1}^{n_2}$; tuning grids for the smoothing parameter and penalties.

1: For each target location $u$, compute kernel weights on the Hamming cube and estimate $\mu_1(u)$, $\mu_2(u)$, and $\Sigma(u)$ by the normalized kernel estimators.

2: Estimate the location-dependent discriminant direction by

$$\widehat{\beta}(u) = \arg\min_{b \in \mathbb{R}^p} \left\{ b^\top \widehat{\Sigma}(u) b - 2b^\top \{\widehat{\mu}_1(u) - \widehat{\mu}_2(u)\} + \lambda_\beta |b|_1 \right\}.$$

3: Estimate the intercept $\eta(u)$ by fitting the sparse logistic model

$$\widehat{\eta}(u) = \widehat{A}_0 + \widehat{A}^\top u.$$

4: Select the smoothing parameter and penalties by cross-validation using the misclassification criterion described in Section 4.1.

5: For a new observation $(Z, u)$, classify it to class 1 if

$$\widehat{\beta}(u)^\top \left\{ Z - \frac{\widehat{\mu}_1(u) + \widehat{\mu}_2(u)}{2} \right\} + \widehat{\eta}(u) > 0,$$

and to class 2 otherwise.

**Output** A classifier for a new observation $(Z, u)$.

some abuse of notations, for a given $\mathbf{u} \in \{0, 1\}^d$, we denote $\Delta_{\mathbf{u}} = \mathbf{U} - \mathbf{u}$ and $N_{\mathbf{u}} = |\mathbf{U} - \mathbf{u}|_1$, where $\mathbf{U}$ is a random variable with probability mass function $p_1(\mathbf{U})$ or $p_2(\mathbf{U})$, depending on whether $\mathbf{U}$ is from class 1 or 2. We use $m(\mathbf{u})$ as a generic notation to denote any of the following conditional mean functions: $E[X_{1i}^k|\mathbf{u}]$, $E[Y_{1i}^k|\mathbf{u}]$, $E[(X_{1i}X_{1j})^k|\mathbf{u}]$, and $E[(Y_{1i}Y_{1j})^2|\mathbf{u}]$ with $k = 1$ or $2$, $1 \leq i, j \leq p$. We denote $a \asymp b$ if $a = O(b)$ and $b = O(a)$. Throughout this paper, $c, C, C_0, C_1, C_2, \ldots$ refer to some generic constants that may take different values in different places.

## 3.1 Concentration inequalities

To study the property of the proposed estimators, we make the following assumptions.

(C1). There exists a constant $B \in (0, 0.5]$ such that $B \leq d^{-1}EN_{\mathbf{u}} \leq 1 - B$ holds for any $\mathbf{u} \in \{0, 1\}^d$. There exist constants $C > 0$ and $0 \leq \alpha < \frac{1}{2}$ such that for any $\mathbf{u} \in \{0, 1\}^d$, $E(\sum_{i=1}^d W_{\mathbf{u}}^{(i)})^k \leq Ck!d^{1+\alpha(k-2)}$ for any $k \geq 2$, where $\mathbf{W}_{\mathbf{u}} = (W_{\mathbf{u}}^{(1)}, \ldots, W_{\mathbf{u}}^{(d)})^\top = \Delta_{\mathbf{u}} - E\Delta_{\mathbf{u}}$.

(C2). Write $n = n_1 + n_2$. We assume that $n_1 \asymp n_2$, $\frac{\log(p+d)}{n} + \frac{\log(p+d)}{d \log^2 n} \to 0$, and for $h = h_x, h_y, h_{xx}$ or $h_{yy}$, there exists a positive constant $C > 0$ such that $\frac{\log(d+n)}{h^2 d} \leq C$.

(C3). For any $\mathbf{u} \in \{0, 1\}^d$ and $t > 0$, denote

$$\kappa_{\mathbf{u}}(t) := \frac{E[m(\mathbf{U}) - m(\mathbf{u})] \exp\{-(dt)^{-1}N_{\mathbf{u}}\}}{E \exp\{-(dt)^{-1}N_{\mathbf{u}}\}},$$

and let $\kappa(t) = \sup_{\mathbf{u} \in \{0,1\}^d} \kappa_{\mathbf{u}}(t)$. We assume that $\kappa(t) \to 0$ as $t \to 0$ and $dt \to \infty$.

(C4). There exists a positive constant $M$ such that $\sup_{1 \le i \le p, \mathbf{u} \in \{0,1\}^d} EX_{1i}^4(\mathbf{u}) \le M < \infty$ and $\sup_{1 \le i \le p, \mathbf{u} \in \{0,1\}^d} EY_{1i}^4(\mathbf{u}) \le M < \infty$. There exists a constant $M_\Sigma > 1$ such that

$$M_\Sigma^{-1} \le \inf_{\mathbf{u} \in \{0,1\}^d} \lambda_1(\Sigma(\mathbf{u})) \le \sup_{\mathbf{u} \in \{0,1\}^d} \lambda_p(\Sigma(\mathbf{u})) \le M_\Sigma,$$

where $\lambda_1(\Sigma(\mathbf{u}))$ and $\lambda_p(\Sigma(\mathbf{u}))$ are the smallest and largest eigenvalues of $\Sigma(\mathbf{u})$, respectively.

Condition (C1) is a regularization condition on the discrete variable $\mathbf{U}$, and is generally true when the success probability of each element in $\mathbf{U}$ is bounded away from zero and one. Condition (C2) specifies the order of the bandwidth $h$. Unlike classical results in kernel smoothing where $h$ is chosen to balance the bias and variance, our $h$ here has to be large enough to ensure the small ball probability $E \exp\{-(hd)^{-1}|\mathbf{U} - \mathbf{u}|_1\}$ is large enough. Specifically, as a form of Bochner's Lemma, a commonly applied result in classical kernel smoothing estimation is that $\frac{1}{h} EK\left(\frac{|\mathbf{U}-\mathbf{u}|_1}{dh}\right)$ would tend to the density function of $\mathbf{u}$ for a properly chosen kernel function under some regular conditions (Bosq, 2012). However, such a conclusion fails in our case. More specifically, suppose we use a continuous density to approximate the discrete probability mass function of $\mathbf{u}$ as $d \to \infty$. With some abuse of notations, let $p_d(\mathbf{u})$ be the point mass probability of $\mathbf{u}$. The approximated density at location $\mathbf{u}$, following traditional arguments, is given as $f(\mathbf{u}) = \lim_{\epsilon \to 0^+} \frac{1}{\epsilon} \sum_{\mathbf{u}_\epsilon : d^{-1}|\mathbf{u}_\epsilon - \mathbf{u}|_1 \le \epsilon} p_d(\mathbf{u}_\epsilon)$. Similar to the small ball probability issue in the Hong and Linton (2016), such a density relies on the choice of $\epsilon$ and hence is not well defined. On the one hand, as $d$ grows, the point mass function at $\mathbf{u}$ converges to zero in an exponential rate. Such a point mass probability cannot be well estimated unless the sample size also grows exponentially in $d$. To tackle this issue, other than evaluating the denominator and numerator in the Nadaraya-Watson type estimators directly, we establish concentration inequalities for their normalized versions such as $\frac{1}{n_1} \sum_{j=1}^{n_1} \frac{\exp\{-(dh_x)^{-1}|\mathbf{U}_j - \mathbf{u}|_1\}}{E \exp\{-(dh_x)^{-1}|\mathbf{U}_1 - \mathbf{u}|_1\}}$ and $\frac{1}{n_1} \sum_{j=1}^{n_1} \frac{\exp\{-(dh_x)^{-1}|\mathbf{U}_j - \mathbf{u}|_1\} X_{ji}}{E \exp\{-(dh_x)^{-1}|\mathbf{U}_1 - \mathbf{u}|_1\}}$. Similar to equation (6) of Hong and Linton (2016), the normalizing coefficient $E \exp\{-(hd)^{-1}|\mathbf{U} - \mathbf{u}|_1\}$ can be viewed as an integrated small ball probability. Condition (C3) quantifies the smoothness of $m(\mathbf{u})$. Condition (C3) plays the same role as a smoothness condition in classical kernel smoothing: it controls the local bias of the kernel estimator. Since the smoothing variable $U$ is a high-dimensional binary vector, a standard fixed-dimensional Hölder or Lipschitz condition is not directly applicable. Instead, (C3) is formulated through the normalized kernel-weighted difference between $m(U)$ and $m(u)$ over the Hamming cube. The following examples illustrate how (C3) can be verified. In particular, Examples 2 and 3 can be viewed as Lipschitz-type sufficient conditions adapted to high-dimensional discrete variables.

**Example 1. Smoothness on the expected different function on the contour**
Note that for any integer $s$, $\mathcal{C}_{\mathbf{u},s} := \{\mathbf{U} : N_\mathbf{u} = s\}$ defines a contour with radius $s$ from the center $\mathbf{u}$. Let $G_\mathbf{u}(s) := E[m(\mathbf{U}) - m(\mathbf{u})|N_\mathbf{u} = s]$ be the expected difference of $m(\cdot)$ over all the $\mathbf{U}$'s on the contour $\mathcal{C}_{\mathbf{u},s}$. (C3) is satisfied if $G_\mathbf{u}(s)$ is smooth in the sense that $\frac{E[G_\mathbf{u}(N_\mathbf{u}) \exp\{-(dt)^{-1} N_\mathbf{u}\}]}{E \exp\{-(dt)^{-1} N_\mathbf{u}\}} = \kappa_\mathbf{u}(t) \to 0$. As an example, suppose for any $\mathbf{u}_i \in \mathcal{C}_{\mathbf{u},1}$, we have $m(\mathbf{u}_i) = m(\mathbf{u}) + s_{\mathbf{u},\mathbf{u}_i}$, where $s_{\mathbf{u},\mathbf{u}_i}$ is the a "signal" generated from $s_{\mathbf{u},\mathbf{u}_i} = I\{\varepsilon_\mathbf{u} = 0\} N(0, \sigma_d^2)$ for some constant $\tau > 0$ and variance $\sigma_d^2 > 0$, and some independent Bernoulli random innovation $\varepsilon_\mathbf{u}$ such that $P(\varepsilon_\mathbf{u} = 0) = P(1 - \varepsilon_\mathbf{u} = 0) = \pi_\mathbf{u}$. The variance $\sigma_d^2$ captures the level of fluctuation of $m(\mathbf{u})$ near the neighborhood locations, and the point mass probability $\pi_\mathbf{u}$ controls the proportion of neighbors

that take the same value as $m(\mathbf{u})$. Under this setting, we have, when $|\mathbf{u}_i - \mathbf{u}|_1 = s$, $m(\mathbf{u}_i) = m(\mathbf{u}) + O_p(\sqrt{s\pi_{\mathbf{u}}}\sigma_d)$. Condition (C3) is satisfied when $\sqrt{\pi_{\mathbf{u}}}\sigma_d \to 0$ fast enough such that $G_{\mathbf{u}}(s) = o(1)$ for all $s = 1, \ldots, d$. This can be true when, for example, $\sqrt{\pi_{\mathbf{u}}}\sigma_d = O(d^{-1/2-\tau})$ for some constant $\tau > 0$. The term $\sqrt{\pi_{\mathbf{u}}}\sigma_d$ in this case captures the order of the difference between $m(\mathbf{u})$ and the expectation $E[m(\mathbf{U})|\mathbf{U} \in \mathcal{C}_{\mathbf{u},1}]$ over the contour $\mathcal{C}_{\mathbf{u},1}$.

**Example 2. Lipschitz with exponential order**

For any $\mathbf{u}_1, \mathbf{u} \in \{0,1\}^d$, suppose there exists a constant $0 \le c \le 1$ such that,

$$|m(\mathbf{u}_1) - m(\mathbf{u})| \le \kappa_1 \exp\left\{\frac{|\mathbf{u}_1 - \mathbf{u}|_1^c - EN_{\mathbf{u}}^c}{(d\log d)^{\frac{c}{2}}}\right\},$$

for some $\kappa_1 \to 0$. Note that by Jensen inequality we have for any $c \in [0,1]$, $EN_{\mathbf{u}}^c \le (EN_{\mathbf{u}})^c$. By Kimball's inequality, Lemma A.1, Conditions (C1) and (C4), there exists a large enough constant $C_1 > 0$ such that

$$
\begin{aligned}
\frac{E[m(\mathbf{U}) - m(\mathbf{u})]\exp\{-(dt)^{-1}N_{\mathbf{u}}\}}{E\exp\{-(dt)^{-1}N_{\mathbf{u}}\}} &\le \kappa_1 E \exp\left\{\frac{N_{\mathbf{u}}^c - EN_{\mathbf{u}}^c}{(d\log d)^{\frac{c}{2}}}\right\} \\
&= \kappa_1 E \exp\left\{\frac{EN_{\mathbf{u}}^c}{(d\log d)^{\frac{c}{2}}} \cdot \left(\frac{N_{\mathbf{u}}^c}{EN_{\mathbf{u}}^r} - 1\right)\right\} \\
&= O\left(\kappa_1 \exp\left\{C_1\left(\frac{d}{\log d}\right)^{\frac{c}{2}} \cdot \left(\frac{\log d}{d}\right)^{\frac{c}{2}}\right\}\right) \\
&= O(\kappa_1).
\end{aligned}
$$

**Example 3. Centered Lipschitz with exponential order**

Existing literature for estimating the distribution of high dimensional discrete variables sometimes models the probability mass as a function of the centered variable instead (Grund and Hall, 1993). We hence consider the following Lipschitz condition centered at the mean of $N_{\mathbf{u}}$: For any $\mathbf{u}_1, \mathbf{u} \in \{0,1\}^d$, there exist constants $c \ge 0$, $C > 0$ such that,

$$|m(\mathbf{u}_1) - m(\mathbf{u})| \le C\kappa_1 \exp\left\{\frac{(|\mathbf{u}_1 - \mathbf{u}|_1 - EN_{\mathbf{u}})^c}{(d\log d)^{\frac{c}{2}}}\right\},$$

for some $\kappa_1 \to 0$. By Kimball's inequality and Lemma A.1 we have,

$$
\begin{aligned}
\frac{E[m(\mathbf{U}) - m(\mathbf{u})]\exp\{-(dt)^{-1}N_{\mathbf{u}}\}}{E\exp\{-(dt)^{-1}N_{\mathbf{u}}\}} &= \frac{E[m(\mathbf{U}) - m(\mathbf{u})]\exp\{-(dt)^{-1}(N_{\mathbf{u}} - EN_{\mathbf{u}})\}}{E\exp\{-(dt)^{-1}(N_{\mathbf{u}} - EN_{\mathbf{u}})\}} \\
&\le C\kappa_1 E \exp\left\{\frac{(N_{\mathbf{u}} - EN_{\mathbf{u}})^r}{(d\log d)^{\frac{r}{2}}}\right\} \\
&= O(\kappa_1).
\end{aligned}
$$

Next we establish concentration inequalities for the weighted estimators of the mean and covariance matrix functions. Note that in practice, the sample size in either the training dataset or the testing dataset can be much smaller than the total locations $2^d$. For simplicity, we shall assume that the region of interest is the ball centered at $\mathbf{v} \in \{0,1\}^d$ with radius $r$: $B_{\mathbf{v}}(r) := \{\mathbf{u} \in \{0,1\}^d : |\mathbf{u} - \mathbf{v}|_1 \le r\}$. Let $\kappa(\cdot)$ be defined as in Condition (C3). The following theorems

establish uniform consistency for any $\mathbf{u}$ in the ball $B_{\mathbf{v}}(r)$.

**Theorem 3.1.** *Under Conditions (C1)-(C4), when $n_1$ and $n_2$ are large enough, there exist constants $C_1 > 0$, $C_2 > 0$ such that for any $\epsilon_n > \kappa(h)$,*

$$P\left(\sup_{1 \le i \le p, \mathbf{u} \in B_{\mathbf{v}}(r)} |\hat{\mu}_{1i}(\mathbf{u}) - \mu_{1i}(\mathbf{u})| > \epsilon_n\right) \le C_1 p d^r \exp\left\{-C_2 n_1 (\epsilon_n - \kappa(h))^2\right\},$$

*and*

$$P\left(\sup_{1 \le i \le p, \mathbf{u} \in B_{\mathbf{v}}(r)} |\hat{\mu}_{2i}(\mathbf{u}) - \mu_{2i}(\mathbf{u})| > \epsilon_n\right) \le C_1 p d^r \exp\left\{-C_2 n_2 (\epsilon_n - \kappa(h))^2\right\}.$$

Similarly, we can show the following.

**Theorem 3.2.** *Under Conditions (C1)-(C4), when $n_1$ and $n_2$ are large enough, there exist constant $C_1 > 0$, $C_2 > 0$ such that for any $\epsilon_n > \kappa(h)$,*

$$P\left(\sup_{\mathbf{u} \in B_{\mathbf{v}}(r)} \left\|\hat{\Sigma}(\mathbf{u}) - \Sigma(\mathbf{u})\right\|_{\infty} > \epsilon_n\right) \le C_1 p^2 d^r \exp\left\{-C_2 n (\epsilon_n - \kappa(h))^2\right\}.$$

Theorems 3.1 and 3.2 above provide concentration results for $\hat{\mu}_1(\mathbf{u})$ and $\hat{\mu}_2(\mathbf{u})$. In particular, the right hand side of the concentration inequalities will tend to 0 when we set $\epsilon_n = C\sqrt{\frac{\log(p+d)}{n}} + \kappa(h)$ for some large enough constant $C > 0$. The rate $\sqrt{\frac{\log(p+d)}{n}}$ echoes a classical rate that quantifies its dependence on the dimension and the sample size, while the term $\kappa(h)$ is a bias caused by local smoothing. It is generally hard to evaluate $\kappa(h)$ unless some strong structural assumptions are imposed for $m(\mathbf{u})$. Under the classical context, the bandwidth $h$ is usually chosen to obtain a trade-off between the bias and the variance, and hence it is theoretically crucial to know the rate of the bias. However, under the setting that $m(\mathbf{u})$ is high dimensional, the $h$ that provides the best bias-variance trade-off may not necessarily provide any guarantee for the uniform convergence of the estimator, which is an essential requirement for establishing consistency under high dimensionality. Alternatively, we suggest setting $h$ to be large enough (as in Condition (C3)) to ensure an analogue of Bochner's Lemma (i.e., Lemma A.2 in the Supplementary) to hold, and as a result, concentration results in the above two theorems can be appropriately established. Practically, although it is common to select the bandwidth by minimizing the mean integrated squared error via cross validation, for classification with high dimensional mixed variables, we have found that it works better to choose $h$ to minimize the misclassification rate, as described in Section 4.

## 3.2  Consistency of $\hat{\beta}(\mathbf{u})$

### 3.2.1  $\ell_1$-penalized estimation

Before we introduce the main results for the estimation of $\beta(\mathbf{u})$, we study the theoretical properties of the solution of the following generic penalized quadratic loss. These results will be used to prove the consistency of $\hat{\beta}(\mathbf{u})$ later. For a given $\mathbf{u} \in \{0, 1\}^d$, let $\hat{\Omega}(\mathbf{u})$ and $\hat{\mathbf{a}}$ be consistent estimators of a $p \times p$ parameter matrix $\Omega(\mathbf{u})$ and a $p$ dimensional parameter $\mathbf{a}$, respectively. For

a given tuning parameter $\lambda$, we define the $l_1$ penalized estimator of $\mathbf{b}^*(\mathbf{u}) := \mathbf{\Omega}(\mathbf{u})^{-1}\mathbf{a}(\mathbf{u})$ as:

$$\hat{\mathbf{b}}(\mathbf{u}) = \underset{\mathbf{b} \in R^p}{\arg\min} \frac{1}{2}\mathbf{b}^\top \hat{\mathbf{\Omega}}(\mathbf{u})\mathbf{b} - \hat{\mathbf{a}}^\top(\mathbf{u})\mathbf{b} + \lambda|\mathbf{b}|_1. \tag{10}$$

Denote the support of $\mathbf{b}^*(\mathbf{u})$ as $\mathcal{S}_\mathbf{u} = \{i : b_i^*(\mathbf{u}) \neq 0\}$ where $b_i^*(\mathbf{u})$ is the $i$-th element of $\mathbf{b}^*(\mathbf{u})$. When there is no ambiguity we shall use $\mathcal{S}$ instead of $\mathcal{S}_\mathbf{u}$ in some occasions. For example, we shall use $\mathbf{b}_\mathcal{S}(\mathbf{u})$ instead of $\mathbf{b}_{\mathcal{S}_\mathbf{u}}(\mathbf{u})$ to denote the nonzero subset of $\mathbf{b}(\mathbf{u})$. The following proposition establishes an upper bound for the estimation error of $\hat{\mathbf{b}}(\mathbf{u})$ in terms of the estimation accuracy of $\hat{\mathbf{a}}(\mathbf{u})$ and $\hat{\mathbf{\Omega}}(\mathbf{u})$.

**Proposition 3.3.** *Denote* $\epsilon_\mathbf{u} = \|\hat{\mathbf{a}}(\mathbf{u}) - \mathbf{a}(\mathbf{u})\|_\infty + \|[\hat{\mathbf{\Omega}}(\mathbf{u}) - \mathbf{\Omega}(\mathbf{u})]\mathbf{b}^*(\mathbf{u})\|_\infty$, *and* $e_\mathbf{u} = \|\mathbf{b}^*(\mathbf{u})\|_0 \|\mathbf{\Omega}_{\mathcal{S},\mathcal{S}}(\mathbf{u})^{-1}\|_L \|\hat{\mathbf{\Omega}}(\mathbf{u}) - \mathbf{\Omega}(\mathbf{u})\|_\infty$. *Assume the following inequalities hold:*

$$\sup_{\mathbf{u} \in B_\mathbf{v}(r)} \{\|\mathbf{\Omega}_{\mathcal{S}^c,\mathcal{S}}(\mathbf{u})\mathbf{\Omega}_{\mathcal{S},\mathcal{S}}(\mathbf{u})^{-1}\|_L + e_\mathbf{u}\} < 1,$$

$$\sup_{\mathbf{u} \in B_\mathbf{v}(r)} 2[1 - \|\mathbf{\Omega}_{\mathcal{S}^c,\mathcal{S}}(\mathbf{u})\mathbf{\Omega}_{\mathcal{S},\mathcal{S}}(\mathbf{u})^{-1}\|_L - 2e_\mathbf{u}]^{-1}\epsilon_\mathbf{u} < \lambda.$$

*We have, for any* $\mathbf{u} \in B_\mathbf{v}(r)$,

*(i)* $\hat{\mathbf{b}}_{\mathcal{S}^c}(\mathbf{u}) = \mathbf{0}$;

*(ii)* $\|\hat{\mathbf{b}}(\mathbf{u}) - \mathbf{b}^*(\mathbf{u})\|_\infty \leq 2(1 - e_\mathbf{u})^{-1}\|\mathbf{\Omega}_{\mathcal{S},\mathcal{S}}(\mathbf{u})^{-1}\|_L \lambda$.

Proposition 3.3 provides a general result for the oracle property of an estimator defined by minimizing the estimated quadratic loss (10). We remark that Proposition 3.3 can be applied to many different statistical problems where quadratic loss taking the form (10) is adopted. In particular, we do not impose any assumption on the signal strength of the nonzero parameters. Conditions in the above proposition rely on the magnitude of $\|\mathbf{\Omega}_{\mathcal{S}^c,\mathcal{S}}(\mathbf{u})\mathbf{\Omega}_{\mathcal{S},\mathcal{S}}(\mathbf{u})^{-1}\|_L$, which is related to the well known irrepresentable condition (Zhao and Yu, 2006; Zou, 2006), and the uniform estimation error bounds of $\|\hat{\mathbf{a}}(\mathbf{u}) - \mathbf{a}(\mathbf{u})\|_\infty$ and $\|\hat{\mathbf{\Omega}}(\mathbf{u}) - \mathbf{\Omega}(\mathbf{u})\|_\infty$, which require specific evaluation for different applications.

### 3.2.2 Oracle properties of $\hat{\beta}(u)$

Next we establish the oracle properties of $\hat{\beta}(u)$ by using the results obtained in the previous subsections. With some abuse of notations, denote the support of $\beta(\mathbf{u})$ as $\mathcal{S}_\mathbf{u} = \{i : \beta_i(\mathbf{u}) \neq 0\}$ where $\beta_i(\mathbf{u})$ is the $i$th element of $\beta(\mathbf{u})$. Similarly, we use $\hat{\mathcal{S}}_\mathbf{u}$ to denote the support of the estimator $\hat{\beta}(\mathbf{u})$ based on (6). From Proposition 3.3 and the uniform bounds established in Section 3.1, we have:

**Theorem 3.4.** *Let Conditions (C1)-(C4) hold. In addition, assume that*

$$\left(\sqrt{\frac{\log(p+d)}{n}} + \kappa(h)\right) \sup_{\mathbf{u} \in B_\mathbf{v}(r)} \|\beta(\mathbf{u})\|_0 \|\Sigma_{\mathcal{S},\mathcal{S}}(\mathbf{u})^{-1}\|_L \to 0,$$

*and there exists a constant* $0 < \kappa_0 < 1$ *such that* $\sup_{\mathbf{u} \in B_\mathbf{v}(r)} \|\Sigma_{\mathcal{S}^c,\mathcal{S}}(\mathbf{u})\Sigma_{\mathcal{S},\mathcal{S}}(\mathbf{u})^{-1}\|_L < 1 - \kappa_0$. *By choosing* $\lambda_\beta = C_2\left(\sqrt{\frac{\log(p+d)}{n}} + \kappa(h)\right) \sup_{\mathbf{u} \in B_\mathbf{v}(r)}(|\beta(\mathbf{u})|_1 + 1)$ *for some large enough constant* $C_2$, *and denoting* $M_r = \sup_{\mathbf{u} \in B_\mathbf{v}(r)} \|\Sigma_{\mathcal{S},\mathcal{S}}(\mathbf{u})^{-1}\|_L$, *we have, for any given* $r = O(\log_d(p+d))$,

*(i)* $P\left(\bigcap_{\mathbf{u}\in B_{\mathbf{v}}(r)}\{\hat{\mathcal{S}}_{\mathbf{u}}=\mathcal{S}(\mathbf{u})\}\right)=1-O((p+d)^{-1})$;

*(ii)* $P\left(\sup_{\mathbf{u}\in B_{\mathbf{v}}(r)}\|\hat{\beta}(\mathbf{u})-\beta(\mathbf{u})\|_{\infty}\le 2\lambda_{\beta}M_r)\right)=1-O((p+d)^{-1})$.

Note that we do allow the support of $\beta(\mathbf{u})$ to be different across different $\mathbf{u}\in B_{\mathbf{v}}(r)$. Part (i) in Theorem 3.4 states that the common support of non-informative features can be consistently identified. Part (ii) of Theorem 3.4 indicates that the estimation error for the nonzero elements is of order $O_p(\lambda_{\beta}M_r)$. This is similar to the error bound obtained in Theorem 2 of Mai et al. (2012). However, instead of imposing any assumption for the minimal signal $|\beta(\mathbf{u})|_{\min}$ as in Condition 2 of Mai et al. (2012), our error bound relies on the total signal strength $|\beta(\mathbf{u})|_1$ through the choice of $\lambda_{\beta}$.

### 3.3 Consistency of $\hat{\eta}(\mathbf{u})$

Theoretical properties of penalized logistic regression have been well explored in the literature; see for example Meier et al. (2008) and Rocha et al. (2009). Other than studying the excess risk or the global error of the estimator $\hat{\eta}$ as in Meier et al. (2008) or establishing consistency for the coefficient estimators with requires additional assumptions on the signal strength for the nonzero parameters, here we directly explore the estimation accuracy of $\hat{\eta}(\mathbf{u})=\hat{A}_0+\hat{\mathbf{A}}^{\top}\mathbf{u}$ in estimating $\eta(\mathbf{u})=A_0+\mathbf{A}^{\top}\mathbf{u}$. We first assume the following conditions hold:

(C5). Let $\mathbf{W}=\mathbf{U}I\{L=1\}+\mathbf{V}I\{L=2\}$ where $\mathbf{U}$ and $\mathbf{V}$ are independent binary vectors with density functions $p_1(\cdot)$ and $p_2(\cdot)$, respectively. We assume that the covariance matrix $\mathbf{H}=\mathrm{Var}(\mathbf{W})$ is positive definite with eigenvalues bounded away from zero.

(C6). Let $\mathcal{M}=\{j:A_j\ne 0\}$ and $M=|\mathbf{A}_1|_0$. We assume that $1-\max_{e\in\mathcal{M}^c}|\mathbf{H}_{e,\mathcal{M}}\mathbf{H}_{\mathcal{M},\mathcal{M}}^{-1}|>0$, and $M^2e^M\sqrt{\frac{\log d}{n}}\to 0$.

Condition (C6) is an irrepresentability assumption to guarantee model selection consistency. The following theorem provides the uniform estimation error for $\hat{\eta}(\mathbf{u})$.

**Theorem 3.5.** *Let Conditions (C5) and (C6) hold. We have*

$$\sup_{\mathbf{u}\in\{0,1\}^d}|\hat{\eta}(\mathbf{u})-\eta(\mathbf{u})|=O_p\left(M^2e^M\sqrt{\frac{\log d}{n}}\right).$$

From Theorem 3.5, the uniform error bound is reduced to $O_p\left(\sqrt{\frac{\log d}{n}}\right)$ when the number of nonzero parameters $M$ is bounded.

### 3.4 Misclassification rate

Given $\mathbf{U}=\mathbf{u}$, denote $D(\mathbf{Z};\mathbf{u})=\beta(\mathbf{u})^{\top}\left[\mathbf{Z}-\frac{\mu_1(\mathbf{u})+\mu_2(\mathbf{u})}{2}\right]+\eta(\mathbf{u})$, and correspondingly denote $\hat{D}(\mathbf{Z};\mathbf{u})=\hat{\beta}(\mathbf{u})^{\top}\left[\mathbf{Z}-\frac{\hat{\mu}_1(\mathbf{u})+\hat{\mu}_2(\mathbf{u})}{2}\right]+\hat{\eta}(\mathbf{u})$. The optimal Bayes' risk is given as

$$R_{\mathbf{u}}=\pi_1R_{\mathbf{u}}(2|1)+\pi_2R_{\mathbf{u}}(1|2),$$

where $R_{\mathbf{u}}(2|1) = P(D(\mathbf{Z};\mathbf{u}) \leq 0 | \mathbf{Z} \in N(\mu_1(\mathbf{u}), \Sigma(\mathbf{u}))$ and $R_{\mathbf{u}}(1|2) = P(D(\mathbf{Z};\mathbf{u}) > 0 | \mathbf{Z} \in N(\mu_2(\mathbf{u}), \Sigma(\mathbf{u}))$. Correspondingly, the misclassification rate of our proposed method is

$$\hat{R}_{\mathbf{u}} = \pi_1 \hat{R}_{\mathbf{u}}(2|1) + \pi_2 \hat{R}_{\mathbf{u}}(1|2),$$

where $\hat{R}_{\mathbf{u}}(2|1) = P(\hat{D}(\mathbf{Z};\mathbf{u}) \leq 0 | \mathbf{Z} \in N(\mu_1(\mathbf{u}), \Sigma(\mathbf{u}))$ and $\hat{R}_{\mathbf{u}}(1|2) = P(\hat{D}(\mathbf{Z}, \mathbf{u}) > 0 | \mathbf{Z} \in N(\mu_2(\mathbf{u}), \Sigma(\mathbf{u}))$. Note that when $\mathbf{Z} \in N(\mu_2(\mathbf{u}), \Sigma(\mathbf{u}))$, we have,

$$D(\mathbf{Z};\mathbf{u}) \sim N\left(\beta(\mathbf{u})^\top[\mu_2(\mathbf{u}) - \mu_1(\mathbf{u})]/2 + \eta(\mathbf{u}), \beta(\mathbf{u})^\top[\mu_1(\mathbf{u}) - \mu_2(\mathbf{u})]\right).$$

Denote $\Delta(\mathbf{u}) := \beta(\mathbf{u})^\top[\mu_1(\mathbf{u}) - \mu_2(\mathbf{u})] = [\mu_1(\mathbf{u}) - \mu_2(\mathbf{u})]^\top \Sigma^{-1}(\mathbf{u})[\mu_1(\mathbf{u}) - \mu_2(\mathbf{u})]$. We assume:

(C7). $\sup_{\mathbf{u}\in\{0,1\}^d} \Delta(\mathbf{u}) \geq \delta$ for some constant $\delta > 0$.

We remark that $\Delta(\mathbf{u})$ captures how far are the (normalized) centers of the two classes away from each other. Condition (C7) ensures that the center of the two class are separable. The following theorem indicates that the estimated semiparametric classification rule $I\{\hat{D}(\mathbf{Z};\mathbf{u}) > 0\}$ is asymptotically optimal.

**Theorem 3.6.** *Let Conditions (C1)-(C7) hold. For a given* $\mathbf{u} \in \{0,1\}^d$ *of interest, let* $s = \sup_{\mathbf{u}\in B_{\mathbf{v}}(r)} \|\beta(\mathbf{u})\|_0$. *We have:*

$$\sup_{\mathbf{u}\in B_{\mathbf{v}}(r)} |\hat{R}_{\mathbf{u}} - R_{\mathbf{u}}| \tag{11}$$
$$= O_p\left(\left(s + \sup_{\mathbf{u}\in B_{\mathbf{v}}(r)} |\beta(\mathbf{u})|_1\right) M_r \left(\sqrt{\frac{\log(p+d)}{n}} + \kappa(h)\right) + M^2 e^M \sqrt{\frac{\log d}{n}}\right).$$

There are two terms on the right hand side of (11). The first term is similar to those for other high dimensional LDA classifiers with continuous data only (Cai and Liu, 2011; Jiang et al., 2020), and is mainly introduced by the estimation of $\beta(\mathbf{u})$. The second term is introduced by the estimation of $\eta(\mathbf{u})$. Although both $p$ and $d$ are allowed to grow exponentially in $n$, the sparsity requirement for the discrete variables seems to be more restricted as we require $M^2 e^M$ to be $o(\sqrt{\frac{n}{\log d}})$.

## 4 Numerical Studies

### 4.1 Selection of tuning parameters

In what follows we will introduce the use of cross validation for determining the bandwidths and the tuning parameters $\lambda_\beta$ and $\lambda_\eta$.

#### 4.1.1 Selection of the bandwidth and $\lambda_\beta$ for the estimation of $\beta(\mathbf{u})$

One approach to choose the bandwidths $h_x, h_y, h_{xx}$ and $h_{yy}$ is to use leave-one-out cross validation as is usually done in kernel smoothing. This bandwidth selection approach does not recognize the constraint in our theory that the bandwidths should be large enough (see Condition C2) to guarantee an analogue of Bochner's Lemma (i.e., Lemma A.2) to hold. As an

alternative, we propose to select the bandwidths and the tuning parameter $\lambda_\beta$ together by minimizing classification error.

For simplicity, we shall use a common bandwidth parameter $h$ for all the bandwidths $h_x, h_y, h_{xx}$ and $h_{yy}$. Suppose we reformulate the weights by introducing $\theta = \frac{\exp\{-(dh)^{-1}\}}{1+\exp\{-(dh)^{-1}\}}$. That is, by writing $\exp\{-(dh)^{-1}t\} = \left(\frac{\theta}{1-\theta}\right)^t$, we have

$$\hat{\mu}_1(\mathbf{u}) = \sum_{j=1}^{n_1} \frac{(\frac{\theta}{1-\theta})^{|\mathbf{U}_j - \mathbf{u}|_1} \mathbf{X}_j}{\sum_{j=1}^{n_1} (\frac{\theta}{1-\theta})^{|\mathbf{U}_j - \mathbf{u}|_1}}, \qquad \hat{\mu}_2(\mathbf{u}) = \sum_{j=1}^{n_2} \frac{(\frac{\theta}{1-\theta})^{|\mathbf{V}_j - \mathbf{u}|_1} \mathbf{Y}_j}{\sum_{j=1}^{n_2} (\frac{\theta}{1-\theta})^{|\mathbf{V}_j - \mathbf{u}|_1}}.$$

Clearly $\theta \in [0, 0.5]$. When $\theta \to \frac{1}{2}$, $\hat{\mu}_{1i}(\mathbf{u})$ and $\hat{\mu}_{2i}(\mathbf{u})$ reduce to the means of all the samples, and when $\theta \to 0$, $\hat{\mu}_{1i}(\mathbf{u})$ and $\hat{\mu}_{2i}(\mathbf{u})$ reduce to the sample means in the cells only. Coincidentally, under this formulation, we found that subject to a normalizing term $(1 - \theta_x)^d$, the denominator is the same as the smoothing estimator for the distribution of a high dimensional and binary random vector in Aitchison and Aitken (1976) and Grund and Hall (1993). We shall be adopting this new formulation for tuning selection, as the parameter $\theta$ is now bounded, which is practically more convenient for tuning.

Note that Proposition 2.1 implies that the estimation of $\beta(\mathbf{u})$ can be independently conducted by minimizes the expected misclassification rate over the class of zero-intercept classifiers. More specifically, given $(\theta, \lambda_\beta)$, let $\hat{\beta}_{-i}(\mathbf{U}_i)$ and $\hat{\beta}_{-i}(\mathbf{V}_i)$ be the estimators obtained using (6) by leaving $(\mathbf{X}_i, \mathbf{U}_i)$ and $(\mathbf{Y}_i, \mathbf{V}_i)$ out, respectively. We choose $(\theta, \lambda_\beta)$ such that the following misclassification rate is minimized:

$$\begin{aligned} R_0(\lambda_\beta) &= \sum_{i=1}^{n_1} I\left\{\hat{\beta}_{-i}(\mathbf{U}_i)^\top \left(\mathbf{X}_i - \frac{\hat{\mu}_{1,-i}(\mathbf{U}_i) + \hat{\mu}_2(\mathbf{U}_i)}{2}\right) \leq 0\right\} \\ &+ \sum_{j=1}^{n_2} I\left\{\hat{\beta}_{-j}(\mathbf{V}_j)^\top \left(\mathbf{Y}_j - \frac{\hat{\mu}_1(\mathbf{V}_j) + \hat{\mu}_{2,-j}(\mathbf{V}_j)}{2}\right) \geq 0\right\}. \end{aligned}$$

### 4.1.2 Selection of $\lambda_\eta$ for the estimation of $\eta(\mathbf{u})$

Given the chosen $(\theta, \lambda_\beta)$, we denote

$$\begin{aligned} \zeta_i &:= \hat{\beta}_{-i}(\mathbf{U}_i)^\top \left(\mathbf{X}_i - \frac{\hat{\mu}_{1,-i}(\mathbf{U}_i) + \hat{\mu}_2(\mathbf{U}_i)}{2}\right), \\ \zeta_{n_1+j} &:= \hat{\beta}_{-j}(\mathbf{V}_j)^\top \left(\mathbf{Y}_j - \frac{\hat{\mu}_1(\mathbf{V}_j) + \hat{\mu}_{2,-j}(\mathbf{V}_j)}{2}\right), \end{aligned}$$

for $i = 1, \ldots, n_1$ and $j = 1, \ldots, n_2$. Note that these values have been computed when determining $\lambda_\beta$ and hence it requires no extra computation burden. Let $(\hat{A}_{0,-i}, \hat{\mathbf{A}}_{-i})$ and $(\hat{A}_{0,-(n_1+j)}, \hat{\mathbf{A}}_{-(n_1+j)})$ be the estimator based on (5) by leaving $\mathbf{U}_i$ and $\mathbf{V}_j$ out, respectively. We then choose $\lambda_\eta$ by minimizing the following misclassification rate:

$$R(\lambda_\eta) = \sum_{i=1}^{n_1} I\left\{\zeta_i + \hat{A}_{0,-i} + \mathbf{U}_i^\top \hat{\mathbf{A}}_{-i} \leq 0\right\} + \sum_{j=1}^{n_2} I\left\{\zeta_{n_1+j} + \hat{A}_{0,-(n_1+j)} + \mathbf{V}_j^\top \hat{\mathbf{A}}_{-(n_1+j)} \geq 0\right\}.$$

## 4.2 Alternative classifiers

For any $\mathbf{u}_1, \mathbf{u}_2 \in \{0,1\}^d$, we use $d(\mathbf{u}_1, \mathbf{u}_2)$ to denote the distance between $\mathbf{u}_1, \mathbf{u}_2$, where $d(\cdot, \cdot) :$ $\{0,1\}^d \times \{0,1\}^d \mapsto \mathbb{R}$ is a distance metric. While the normalized Hamming distance: $d(\mathbf{u}_1, \mathbf{u}_2) = \frac{|\mathbf{u}_1 - \mathbf{u}_2|_1}{d}$ has been adopted (c.f. Section 2.2) in our estimation of the semiparametric location model, we can also assume that there exists an embedding of the categorical variables in $\mathbb{R}$, i.e, $\mathcal{E}(\cdot) : \{0,1\}^d \mapsto \mathbb{R}$, such that $d(\mathbf{u}_1, \mathbf{u}_2) = |\mathcal{E}(\mathbf{u}_1) - \mathcal{E}(\mathbf{u}_2)|$. In particular, in this numerical study, we will consider the following two cases:

- Linear Embedding: $\mathcal{E}(\mathbf{u}) = \eta(\mathbf{u}) = A_0 + \mathbf{A}^\top \mathbf{u}$, where $\eta(\mathbf{u})$ is optimal linear classifier defined as in (4). As a consequence, we have $d(\mathbf{u}_1, \mathbf{u}_2) = |\mathbf{A}^\top (\mathbf{u}_1 - \mathbf{u}_2)|$. We shall denote the semiparametric location model with such a linear embedding on the categorical variables as $\text{SLM}_{\text{LE}}$.

- Deep Embedding: Embedding of categorical variables via deep neural networks has been well studied in the literature of Natural Language Processing, and one of the most popular methods is the neural network-based model "Word2Vec" Mikolov et al. (2013). While "Word2Vec" was developed for word vectors, Wen et al. (2016) proposed a new model "Cat2Vec" which extended the deep embedding framework of "Word2Vec" to deal with general categorical variables. We shall adopt the "Cat2Vec" approach and maps the high dimensional categorical variables onto the real space $\mathbb{R}$, and we denote the semiparametric location model with such a deep embedding on the categorical variables as $\text{SLM}_{\text{DE}}$.

## 4.3 Simulation

Let $\mathbf{u} = (u_1, \ldots, u_d)^\top$ be a generic location in $\{0,1\}^d$. Given a location $\mathbf{u}$, observations from class 1 are generated from $N(\mu_1(\mathbf{u}), \Sigma(\mathbf{u}))$, and those from class 2 are generated from $N(\mu_2(\mathbf{u}), \Sigma(\mathbf{u}))$. In this simulation, the mean functions $\mu_1(\mathbf{u}) = (\mu_{11}(\mathbf{u}), \cdots, \mu_{1p}(\mathbf{u}))^\top$ and $\mu_2(\mathbf{u}) = (\mu_{21}(\mathbf{u}), \cdots, \mu_{2p}(\mathbf{u}))^\top$ are set as $\mu_1(\mathbf{u}) = \frac{\Sigma(\mathbf{u})\beta(\mathbf{u})}{2} = -\mu_2(\mathbf{u})$. In the following we consider different settings for $\Sigma(\mathbf{u})$ and $\beta(\mathbf{u})$:

**Model 1.** We set the $(i,j)$th element of $\Sigma(\mathbf{u})$ as $\sigma_{ij}(\mathbf{u}) = \bar{u}^{|i-j|}, i, j \in 1, \cdots, p$, where $\bar{u} = d^{-1} \sum_{k=1}^d u_k$, and let $\beta_1(\mathbf{u}) = \beta_2(\mathbf{u}) = 5\left(\frac{\sum_{k=1}^d u_k}{\sqrt{d}} - \frac{\sqrt{d}}{2}\right), \beta_3(\mathbf{u}) = \cdots = \beta_p(\mathbf{u}) = 0$.

**Model 2.** We set the $(i,j)$th element of $\Sigma(\mathbf{u})$ as $\sigma_{ij}(\mathbf{u}) = \bar{u}^{\frac{|i-j|}{2}}, i, j \in 1, \cdots, p$, where $\bar{u} = d^{-1} \sum_{k=1}^d u_k$, and let $\beta_1(\mathbf{u}) = \beta_2(\mathbf{u}) = 5\left(\frac{\sum_{k=1}^d u_k}{\sqrt{d}} - \frac{\sqrt{d}}{2}\right)$, and $\beta_3(\mathbf{u}) = \cdots = \beta_p(\mathbf{u}) = 0$.

**Model 3.** We set the $(i,j)$th element of $\Sigma(\mathbf{u})$ as $\sigma_{ij}(\mathbf{u}) = [3\bar{u}(1 - \bar{u})]^{|i-j|}, i, j \in 1, \cdots, p$, where $\bar{u} = d^{-1} \sum_{k=1}^d u_k$, and let $\beta_1(\mathbf{u}) = \beta_2(\mathbf{u}) = \beta_3(\mathbf{u}) = 5\left(\frac{\sum_{k=1}^d u_k}{\sqrt{d}} - \frac{\sqrt{d}}{2}\right), \beta_4(\mathbf{u}) = \cdots = \beta_p(\mathbf{u}) = 0$.

**Model 4.** We set the $(i,j)$th element of $\Sigma(\mathbf{u})$ as $\sigma_{ij}(\mathbf{u}) = \frac{\bar{u}^{|i-j|}}{\exp\{\bar{u}|i-j|\}}, i, j \in 1, \cdots, p$, where $\bar{u} = d^{-1} \sum_{k=1}^d u_k$, and let $\beta_1(\mathbf{u}) = \cdots = \beta_5(\mathbf{u}) = \text{sign}\left(\frac{\sum_{k=1}^d u_k}{\sqrt{d}} - \frac{\sqrt{d}}{2}\right)\frac{1}{2}\exp\left\{|\frac{2\sum_{k=1}^d u_k}{\sqrt{d}} - \sqrt{d}|\right\}$, and $\beta_6(\mathbf{u}) = \cdots = \beta_p(\mathbf{u}) = 0$.

The locations $\mathbf{V}_i = (V_{i1}, \ldots, V_{id})^\top$ in class 2 are randomly generated as $P(V_{ij} = 0) = 0.5, i = 1, 2, \cdots, n_2, j = 1, 2, \cdots, d$. For class 1, we generate $\mathbf{U}_i = (U_{i1}, \ldots, U_{id})^\top$ as $P(U_{ij} = 1) = 0.5 + \xi_j$ for $i = 1, 2, \cdots, n_1, j = 1, 2, 3, 4, 5$. We simply set $\xi_1 = \cdots = \xi_5 = 0.25$ and $\xi_6 = \cdots = \xi_d = 0$ for Model 1, and set $\xi_1 = \cdots = \xi_5 = 0.3, \xi_6 = \cdots = \xi_d = 0$ for Models 2-4.

We use SLM to denote our proposed semiparametric location model. For comparison, we also consider the following classifiers:

- PLG: $l_1$ Penalized Logistic Regression (Meier et al., 2008).

- PLG$_{\text{inter}}$: $l_1$ Penalized Logistic Regression with the interaction variables $\{X_i U_j : 1 \leq i \leq p, 1 \leq j \leq d\}$.

- RF: Random Forest (Breiman, 2001).

- DSDA: Direct Sparse Discriminant Analysis in Mai et al. (2012).

- FAMD: Factor analysis of mixed data. The mixed variables are first embedded into a low-dimensional continuous representation by factor analysis of mixed data, and a standard classifier is then fitted using the extracted components.

- GB: Gradient boosting classifier. We apply a tree-based gradient boosting classifier to the combined feature matrix after the categorical variables are encoded as binary variables.

- MNB: Mixed-type naive Bayes. This classifier uses Gaussian likelihoods for the continuous variables and Bernoulli or multinomial likelihoods for the categorical variables, together with a conditional independence assumption given the class label.

We first fix the sample size to be $n_1 = n_2 = 200$, and compare all methods listed above under the following dimensions: $(d, p) = (10, 20), (20, 10), (50, 10), (20, 50)$ and $(50, 100)$.

Table 1 reports the mean and standard deviation of the misclassification rates over 200 replications. The best empirical result in each configuration is highlighted in boldface. Overall, SLM is competitive across the four simulation models and often attains the smallest empirical misclassification rate, with performance generally close to the Bayes risk. However, SLM is not uniformly the best method in every configuration. For example, RF and GB achieve smaller errors in some settings. The results should therefore be interpreted as evidence that SLM is a competitive classifier for high-dimensional mixed variables under the location-model settings considered here, rather than as evidence of uniform dominance over all competing methods.

In the second simulation, we fix the dimensions $(d, p)$, and let the sample size $n = n_1 + n_2$ increase from 200 to 500 (with $n_1 = n_2$). The regret, which is defined as the misclassification rate minus the Bayes risk, was computed for each $n$. Figure 1 shows that as $n$ increases, the regret becomes smaller for all the four methods, while the curves for our semiparametric location model generally produce small regret values among all methods as $n$ increases.

## 4.4 Real data analysis

In this section, we investigate the performance of the proposed SLM model by analyzing seven real datasets. We compare SLM to the other classifiers used in simulaton. In addition, we compute the misclassification rates for Classification Tree (CT), which to some degree can be viewed as a non-ensemble version of RF.

The real cases we studied include: Hepatocellular Carcinoma data, Cylinder bands data, Heart-Disease data, Australian credit card application data, Hepatitis data and German Credit

Table 1: The mean and sd of the misclassification rates of SLM, $\text{SLM}_{\text{LE}}$, $\text{SLM}_{\text{DE}}$, PLG, $\text{PLG}_{\text{inter}}$, RF, DSDA, FAMD, GB and MNB over 200 replications under Models 1–4, with $n_1 = n_2 = 200$. The best empirical result in each configuration is highlighted in boldface.

| | Model 1 | | | | | Model 2 | | | | |
|---|---|---|---|---|---|---|---|---|---|---|
| $(d, p)$ | $(10, 20)$ | $(20, 10)$ | $(50, 10)$ | $(20, 50)$ | $(50, 100)$ | $(10, 20)$ | $(20, 10)$ | $(50, 10)$ | $(20, 50)$ | $(50, 100)$ |
| Bayes Risk | 0.158 | 0.176 | 0.192 | 0.177 | 0.191 | 0.147 | 0.166 | 0.18 | 0.172 | 0.189 |
| $R_{\text{SLM}}$ | **0.204(0.020)** | **0.234(0.023)** | 0.278(0.026) | **0.248(0.023)** | **0.279(0.025)** | **0.201(0.023)** | **0.233(0.021)** | 0.278(0.029) | **0.248(0.024)** | **0.278(0.028)** |
| $R_{\text{SLM}_{\text{LE}}}$ | 0.221(0.026) | 0.273(0.023) | 0.305(0.026) | 0.278(0.026) | 0.304(0.024) | 0.213(0.025) | 0.275(0.027) | 0.307(0.026) | 0.276(0.026) | 0.305(0.025) |
| $R_{\text{SLM}_{\text{DE}}}$ | 0.266(0.025) | 0.283(0.022) | 0.305(0.025) | 0.286(0.024) | 0.305(0.026) | 0.260(0.025) | 0.286(0.025) | 0.308(0.026) | 0.283(0.024) | 0.305(0.026) |
| $R_{\text{PLG}_{\text{inter}}}$ | 0.323(0.029) | 0.378(0.037) | 0.455(0.030) | 0.381(0.036) | 0.452(0.030) | 0.321(0.031) | 0.369(0.035) | 0.457(0.030) | 0.376(0.034) | 0.448(0.027) |
| $R_{\text{PLG}}$ | 0.218(0.023) | 0.252(0.023) | 0.280(0.025) | 0.258(0.026) | 0.288(0.026) | 0.217(0.022) | 0.252(0.025) | 0.281(0.025) | 0.257(0.026) | 0.279(0.024) |
| $R_{\text{RF}}$ | 0.232(0.024) | 0.243(0.024) | **0.273(0.025)** | 0.287(0.024) | 0.325(0.027) | 0.235(0.025) | 0.242(0.024) | **0.270(0.026)** | 0.292(0.026) | 0.334(0.029) |
| $R_{\text{DSDA}}$ | 0.217(0.022) | 0.247(0.023) | 0.274(0.023) | 0.262(0.023) | 0.292(0.024) | 0.217(0.022) | 0.247(0.023) | 0.274(0.023) | 0.255(0.023) | 0.287(0.025) |
| $R_{\text{FAMD}}$ | 0.224(0.022) | 0.267(0.026) | 0.338(0.030) | 0.366(0.029) | 0.426(0.026) | 0.222(0.023) | 0.264(0.026) | 0.336(0.028) | 0.344(0.030) | 0.427(0.024) |
| $R_{\text{GB}}$ | 0.222(0.022) | 0.246(0.024) | 0.292(0.022) | 0.278(0.027) | 0.316(0.024) | 0.219(0.025) | 0.236(0.025) | 0.280(0.024) | 0.270(0.023) | 0.315(0.025) |
| $R_{\text{MNB}}$ | 0.282(0.027) | 0.297(0.026) | 0.319(0.023) | 0.311(0.025) | 0.348(0.023) | 0.296(0.022) | 0.324(0.026) | 0.346(0.028) | 0.338(0.023) | 0.369(0.024) |
| | Model 3 | | | | | Model 4 | | | | |
| $(d, p)$ | $(10, 20)$ | $(20, 10)$ | $(50, 10)$ | $(20, 50)$ | $(50, 100)$ | $(10, 20)$ | $(20, 10)$ | $(50, 10)$ | $(20, 50)$ | $(50, 100)$ |
| Bayes Risk | 0.111 | 0.117 | 0.126 | 0.119 | 0.133 | 0.145 | 0.164 | 0.152 | 0.173 | 0.166 |
| $R_{\text{SLM}}$ | **0.182(0.021)** | 0.233(0.026) | 0.280(0.026) | **0.246(0.025)** | **0.259(0.027)** | **0.225(0.022)** | 0.250(0.024) | 0.290(0.029) | **0.261(0.025)** | **0.244(0.027)** |
| $R_{\text{SLM}_{\text{LE}}}$ | 0.213(0.031) | 0.287(0.026) | 0.305(0.025) | 0.290(0.024) | 0.304(0.024) | 0.256(0.026) | 0.289(0.024) | 0.308(0.026) | 0.292(0.024) | 0.308(0.030) |
| $R_{\text{SLM}_{\text{DE}}}$ | 0.287(0.023) | 0.291(0.025) | 0.305(0.025) | 0.294(0.024) | 0.304(0.024) | 0.285(0.023) | 0.293(0.025) | 0.308(0.026) | 0.294(0.024) | 0.308(0.030) |
| $R_{\text{PLG}_{\text{inter}}}$ | 0.326(0.030) | 0.373(0.033) | 0.443(0.042) | 0.371(0.036) | 0.443(0.044) | 0.362(0.037) | 0.410(0.035) | 0.476(0.027) | 0.419(0.029) | 0.474(0.027) |
| $R_{\text{PLG}}$ | 0.220(0.023) | 0.265(0.025) | 0.288(0.025) | 0.269(0.026) | 0.285(0.026) | 0.244(0.023) | 0.265(0.023) | 0.294(0.027) | 0.267(0.028) | 0.287(0.028) |
| $R_{\text{RF}}$ | 0.202(0.022) | 0.222(0.026) | **0.257(0.024)** | 0.277(0.028) | 0.334(0.031) | 0.238(0.024) | **0.237(0.022)** | **0.269(0.025)** | 0.281(0.027) | 0.333(0.031) |
| $R_{\text{DSDA}}$ | 0.216(0.022) | 0.259(0.024) | 0.278(0.024) | 0.268(0.024) | 0.290(0.025) | 0.244(0.023) | 0.260(0.023) | 0.285(0.023) | 0.271(0.025) | 0.302(0.028) |
| $R_{\text{FAMD}}$ | 0.226(0.029) | 0.280(0.025) | 0.344(0.028) | 0.376(0.033) | 0.453(0.025) | 0.260(0.029) | 0.274(0.025) | 0.355(0.027) | 0.370(0.028) | 0.431(0.025) |
| $R_{\text{GB}}$ | 0.196(0.018) | **0.217(0.025)** | 0.274(0.021) | 0.267(0.025) | 0.319(0.027) | 0.236(0.022) | 0.249(0.024) | 0.289(0.024) | 0.280(0.024) | 0.330(0.022) |
| $R_{\text{MNB}}$ | 0.294(0.029) | 0.323(0.031) | 0.334(0.033) | 0.331(0.025) | 0.374(0.029) | 0.285(0.024) | 0.295(0.026) | 0.326(0.027) | 0.319(0.024) | 0.359(0.028) |

data. All these datasets are publicly available on the UCI Machine Learning Repository or the public data platform Kaggle. All categorical variables are translated into binary variables using dummy variable encoding. Missingness in the categorical variable is treated as one category, and mean imputation is used for missing values of the continuous variables. We perform a 10-fold cross-validation and the average misclassification rates are reported in Table 2. The datasets we considered are described as follows:

**Hepatocellular Carcinoma dataset** Hepatocellular carcinoma (HCC) is the most common type of primary liver cancer in adults and the third leading cause of cancer-related death worldwide. This dataset was collected at a University Hospital in Portugal. It contains real clinical data of 165 patients diagnosed with HCC, in which only 102 patients finally survived. There are 22 continuous variables and 59 binary variables.

**Breast Cancer Gene Expression Profiles (METABRIC)** Breast cancer is the most frequent cancer among women, and one of the leading causes of cancer deaths in females. This dataset comes from the Molecular Taxonomy of Breast Cancer International Consortium (META BRIC) database. The original data was published on Nature Communications (Pereira et al., 2016). There are 1904 patients with breast cancer in this data. 489 mRNA Z-scores for 331 genes, and indicators of mutation for 173 genes are recorded. To reduce the computational costs, following Cai and Liu (2011), only 100 mRNA Z-scores with the largest absolute values of the two sample t statistics are used.

**Cylinder bands data** Cylinder bands data contains 20 categorical variables which were transformed into 465 binary variables via dummy variable encoding. There are 17 continuous variables, but we have removed variables "ESA Amperage" and "chrome content", owing to the fact that more than 95% of the observations are taking a same value or having a missing value for these two variables. This dataset contains 277 instances.

**Heart-Disease data** The heart-disease dataset contains 13 attributes (which have been extracted from a larger set of 75). There are 7 categorical variables which can be transformed into 12 binary variables and 6 continuous variables. This dataset contains 270 instances. And there

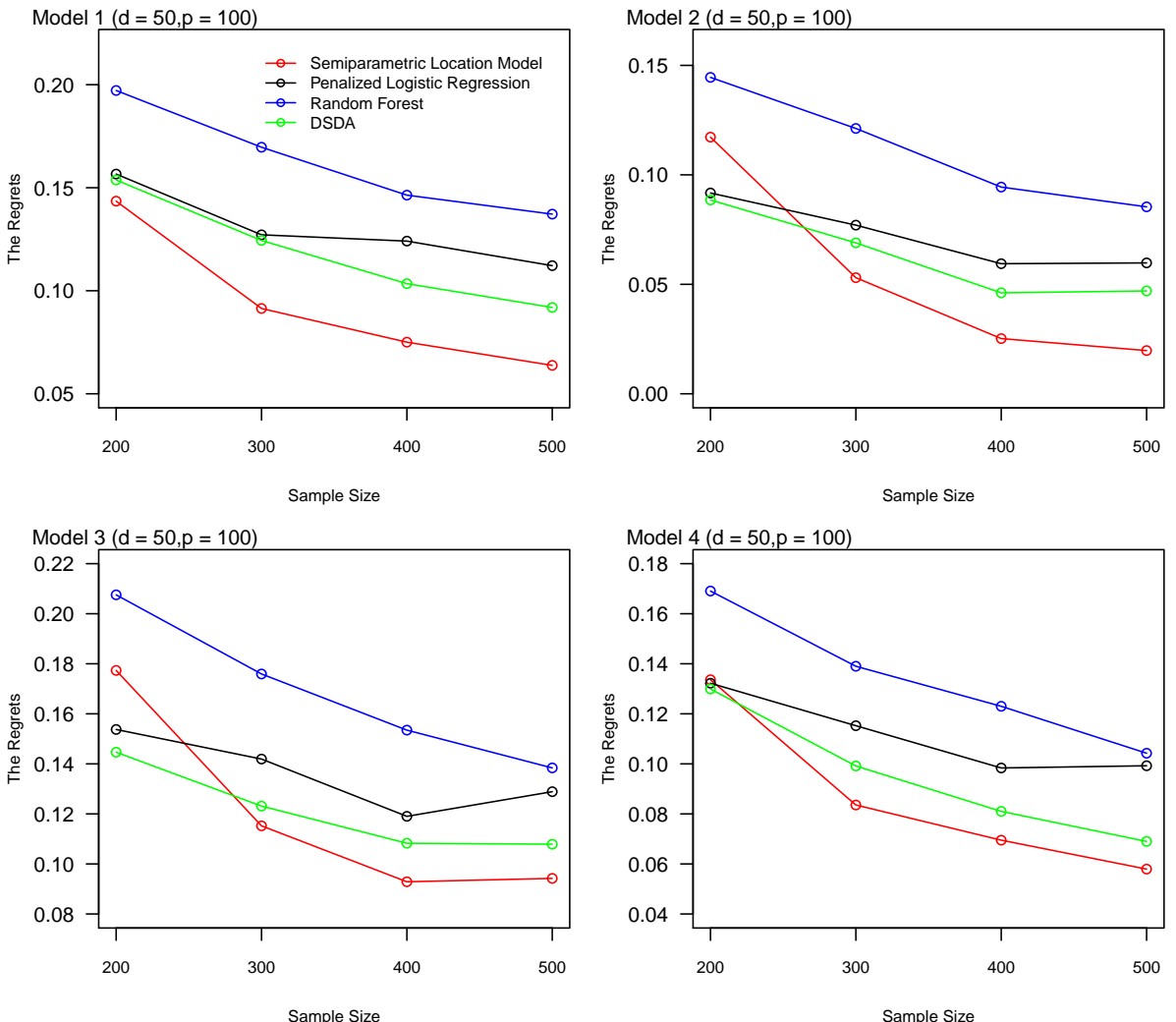

Figure 1: The regrets of SLM,PLG,RF,DSDA under Model 1-4

are no missing values in this dataset.

**Hepatitis data** Hepatitis dataset contains 155 patients diagnosed with hepatitis, in which 123 patients are lived. Missingness in the categorical variable of this dataset is treated as one category. Overall, this dataset contains 7 continuous variables, and 12 categorical variables which were transformed into 22 binary variables.

**Australian Credit Card Approval (ACA) data** This dataset concerns credit card applications. There are 6 numerical variables, and 8 categorical attributes which were transformed into 28 binary variables via dummy variable encoding. This dataset contains 690 instances.

**German Credit data** In this German Credit dataset, there are 7 continuous variables such as duration in month and credit amount, and 13 categorical variables which were transformed into 41 binary variables. The objective is to class a customer as a "good" or "bad" customer. This dataset contains 1000 instances.

From Table 2, we observe that SLM and its variants are competitive across the seven real data sets. SLM achieves the smallest error on HCC, Hepatitis and German Credit; $SLM_{LE}$ performs best on the Cylinder Bands data; and $SLM_{DE}$ performs best on the Heart-Disease data. Among the competing methods, GB gives the smallest error on the Breast Cancer data,

|      | HCC | BC | Bands | Heart | Hepatitis | ACA | German |
|------|-----|-----|-------|-------|-----------|-----|--------|
| SLM | **0.231** | 0.357 | 0.246 | 0.148 | **0.135** | 0.136 | **0.236** |
| SLM$_{\text{LE}}$ | 0.236 | 0.360 | **0.220** | 0.156 | 0.148 | 0.139 | 0.242 |
| SLM$_{\text{DE}}$ | 0.242 | 0.369 | 0.289 | **0.130** | 0.142 | 0.217 | 0.290 |
| PLG$_{\text{inter}}$ | 0.401 | 0.404 | 0.321 | 0.241 | 0.155 | 0.248 | 0.292 |
| PLG | 0.316 | 0.388 | 0.318 | 0.177 | 0.160 | 0.181 | 0.270 |
| RF | 0.303 | 0.371 | 0.231 | 0.156 | 0.141 | **0.129** | 0.251 |
| DSDA | 0.249 | 0.365 | 0.237 | 0.152 | 0.174 | 0.148 | 0.257 |
| CT | 0.376 | 0.420 | 0.310 | 0.181 | 0.238 | 0.139 | 0.283 |
| FAMD | 0.296 | 0.357 | 0.306 | 0.181 | 0.136 | 0.168 | 0.275 |
| GB | 0.241 | **0.353** | 0.227 | 0.178 | 0.148 | 0.133 | 0.240 |
| MNB | 0.321 | 0.387 | 0.321 | 0.163 | 0.169 | 0.230 | 0.242 |

Table 2: Classification errors for real data study under 10-fold cross-validation. The best classifier for each data set is highlighted in boldface.

and RF gives the smallest error on the ACA data. These results suggest that no single method uniformly dominates all data sets, while the proposed SLM family performs competitively across a range of mixed-variable applications.

## 5 Discussion

In this paper we have proposed a semiparametric model based on the optimal Bayes rule for classifying data with high dimensional mixed variables. The oracle Bayes rule essentially relies on the joint distribution of $(\mathbf{X}, \mathbf{U})$ in each class. In our approach, the joint distribution is formulated via the distributions of $\mathbf{X}|\mathbf{U}$ and $\mathbf{U}$, where the discrete variable $\mathbf{U}$ is viewed as a random "location" in $\{0,1\}^d$, and $\mathbf{X}|\mathbf{U}$ is assumed to be Gaussian as in classical LDA, with the mean and covariance matrix defined as functions of the location. We have shown that under this location model, the classification direction and the intercept are separable. To address the curse of dimensionality, we consider a general formulation where the classification direction is assumed to be varying smoothly over the locations, and the intercept is approximated via a first order (linear) expansion. The estimation methods we have adopted in this paper have covered (low order) parametric approximation, and nonparametric smoothing. Different combinations of the parametric and nonparametric approaches could result in different theories and performance in different applications. For example, for the illustrative example in the Supplementary, the optimal classifier can be achieved by adopting a first order linear form for both $\beta(\mathbf{u})$ and $\eta(\mathbf{u})$.

For categorical variables with more than two levels, the binary representation should depend on the nature of the variable. For nominal variables, we use ordinary dummy variable encoding. If the original variables have $K_1, \ldots, K_q$ levels, the resulting binary dimension is $d_{\text{bin}} = \sum_{j=1}^{q}(K_j - 1)$, up to the convention for reference levels and missing categories. The expanded dimension enters the Hamming distance used in the kernel weights, the integrated small-ball probability in the normalization, and the sparsity level of the logistic approximation for $\eta(u)$. For ordinal variables, ordinary dummy encoding is less desirable because it ignores the ordering of the levels. In this case, one can use a cumulative binary encoding: for an ordinal

variable with levels $1, \ldots, K$,

$$1 \mapsto (0, \ldots, 0), \quad 2 \mapsto (1, 0, \ldots, 0), \quad \ldots, \quad K \mapsto (1, \ldots, 1).$$

This encoding preserves the order in the sense that, if $c(r)$ denotes the binary vector corresponding to level $r$, then $\|c(r) - c(s)\|_1 = |r - s|$. The proposed kernel-smoothing framework can then be applied to the expanded binary vector, with the regularity conditions in Section 3 interpreted for this representation.

Above all, the key message we hope to convey in this paper is that categorical variables and continuous variables should be treated differently, and more dedicated modeling strategies should be considered in the presence of other complex structures such as high dimensionality. Apart from the semiparametric classifier we have developed, there are other possible variations that we can consider. First, we can impose different structures for $\beta(\mathbf{u})$ and $\eta(\mathbf{u})$. For example, similar to $\eta(\mathbf{u})$, we can also adopt a linear approximation for the classification direction $\beta(\mathbf{u})$ and the mean and covariance functions. The resulting classifier will reduce to a classifier with linear effects $\mathbf{X}$, $\mathbf{U}$ and their interactions. Second, we can also consider relaxing the Gaussian assumption on $\mathbf{X}$ to a copula model as in Jiang and Leng (2016) and Mai and Zou (2015). Third, it is interesting to apply the idea of the location model to develop ensemble classifiers using for example random subspace for high dimensional data with mixed variables (Tian and Feng, 2023). Last but not least, the proposed idea can be extended to handle the case where $\mathbf{X}$ admits a matrix or tensor structure (Pan et al., 2019). We leave these for future exploration.

## 6 Acknowledgment

Cheng Wang's research is partially supported by NSFC (72495121, 12571288) and the fundamental research funds for the central universities.

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

# A   Technical Lemmas and Proofs

## A.1   An illustrative example

Consider a two-class classification problem with one binary covariate $U$ and one continuous covariate $X$. Assume that the prior for the class label $L$ is balanced, i.e., $P(L = 1) = P(L = 2) = 0.5$, and let $P(U = 0) = P(U = 1) = 0.5$ for both classes. For class 1, suppose $X \sim N(-1, 1)$ under category $U = 0$ and $X \sim N(1, 1)$ under category $U = 1$. For class 2, assume that $X \sim N(1, 1)$ under category $U = 0$ and $X \sim N(-1, 1)$ under category $U = 1$. We have:

(1) The optimal Bayes risk is $\Phi(-1)$ where $\Phi$ is the cumulative distribution function of the standard normal distribution.

(2) The misclassification rate for the optimal linear classifier is $\frac{1}{2}\left[\Phi(-1) + \frac{1}{2}\right]$.

*Proof.* (1). With some simple calculations it can be shown that the optimal Bayes classifier is: classify $(X, U)$ into class 1 if $XU > 0$, and into class 2 otherwise. The corresponding misclassification rate can be easily established.

(2). Without loss of generality, consider a classifier that classify $(X, U)$ into class 1 if and only if $X + aU > b$ for some $a$ and $b$. Let $Z$ be a random variable following the standard normal distribution. The misclassification rate of the linear classifier can be computed as:

$$
\begin{aligned}
R(a, b) \ = \ & \frac{1}{4}P(X + aU > b | L = 2, U = 0) + \frac{1}{4}P(X + aU > b | L = 2, U = 1) \\
& + \frac{1}{4}P(X + aU < b | L = 1, U = 0) + \frac{1}{4}P(X + aU < b | L = 1, U = 1).
\end{aligned}
$$

Note that $P(X + aU > b | L = 2, U = 0) = P(X > b | X \sim N(1, 1)) = P(Z > b - 1)$. Similarly, we have $P(X + aU > b | L = 2, U = 1) = P(Z > b - a + 1)$, $P(X + aU < b | L = 1, U = 0) = P(Z < b + 1)$ and $P(X + aU < b | L = 1, U = 1) = P(Z < b - a - 1)$. Consequently, we have

$$
\begin{aligned}
R(a, b) \ = \ & \frac{1}{2} + \frac{1}{4}P(b - 1 < Z < b + 1) - \frac{1}{4}P(b - a - 1 < Z < b - a + 1) \\
\geq \ & \frac{1}{2} - \frac{1}{4}P(-1 < Z < 1) \\
= \ & \frac{1}{4} + \frac{1}{2}\Phi(-1),
\end{aligned}
$$

where the equal sign in the second inequality is obtained when $a = b \to \infty$.

$\square$

## A.2 Motivation for model (4)

Recall that $\eta(\mathbf{u}) = \frac{\pi_1 p_1(\mathbf{u})}{\pi_2 p_2(\mathbf{u})}$. Note that for any injective function $G(\mathbf{u}) : \{0,1\}^d \mapsto R$, it can be easily shown by induction that $G(\mathbf{u})$ can be written as a polynomial of $\mathbf{u} = (u_1, \ldots, u_d) \in \{0,1\}^d$. Specifically, there exist constants $a_0; a_1, \ldots, a_p; a_{1,2}, a_{1,3}, \ldots, a_{p-1,p}; \ldots; a_{1,2\ldots,d}$, such that

$$G(\mathbf{u}) = a_0 + \sum_{i=1}^d a_i u_i + \sum_{1 \leq i < j \leq d} a_{i,j} u_i u_j + \cdots + a_{1,2\ldots,d} u_1 u_2 \cdots u_d.$$

Suppose $\pi_1 \log p_1(\mathbf{u})$ and $\pi_2 \log p_2(\mathbf{u})$ are injective functions such that

$$\log \pi_j p_j(\mathbf{u}) \tag{12}$$
$$= a_0^{(j)} + \sum_{i=1}^d a_i^{(j)} u_i + \sum_{1 \leq i < j \leq d} a_{i,j} u_i u_j + \cdots + a_{1,2\ldots,d} u_1 u_2 \cdots u_d, \quad j = 1, 2,$$

where the intercepts and first order coefficients between the two classes, $a_0^{(1)}, a_1^{(1)}, \ldots, a_p^{(1)}$ and $a_0^{(2)}, a_1^{(2)}, \ldots, a_p^{(2)}$, are potentially different, and the higher order coefficients $a_{1,2}, \ldots, a_{1,2\ldots,d}$, are assumed to be the same between the two classes. Correspondingly, we have

$$\eta(\mathbf{u}) = \log \frac{\pi_1 p_1(\mathbf{u})}{\pi_2 p_2(\mathbf{u})} = A_0 + \sum_{i=1}^d A_i u_i, \tag{13}$$

with $A_i = a_i^{(1)} - a_i^{(2)}, i = 0, \ldots, d$. The assumption of common higher order coefficients between the two classes in (12) is analogous to the common covariance assumption in the classical LDA setting. Importantly, since all the $u_i$'s are binary variables, the probability of a higher order term $u_{i_1} u_{i_2} \cdots u_{i_k}$ taking the value 1 tends to zero in a geometric rate. Hence the expansion (12) can in general be well approximated by the lower order terms in practice. This is in contrast to the unusual lower order approximation widely applied in linear models where the rational is to achieve certain amount of parsimony. In particular, (13) is true if we only consider model (12) with the first two terms only.

## A.3 Proofs of Proposition 2.1

*Proof.* Given the classification direction $\mathbf{b}(\mathbf{u})$, to minimize (8), it can be shown after some simple calculation that the optimal intercept is given as

$$\tilde{b}_0(\mathbf{u}) = \frac{\mathbf{b}(\mathbf{u})^\top \Sigma(\mathbf{u}) \mathbf{b}(\mathbf{u})}{[\mu_1(\mathbf{u}) - \mu_2(\mathbf{u})]^\top \mathbf{b}(\mathbf{u})} \log \left( \frac{\pi_1 p_1(\mathbf{u})}{\pi_2 p_2(\mathbf{u})} \right).$$

Plugging the above equation into (8), we obtain that, to minimize the expected misclassification rate, it is equivalent to find $\mathbf{b}(\mathbf{u})$ that minimizes

$$R(D(\mathbf{b}(\mathbf{u}), \tilde{b}_0(\mathbf{u})) \tag{14}$$
$$= \sum_{\mathbf{u} \in \{0,1\}^d} \left[ \pi_1 p_1(\mathbf{u}) \Phi \left( -\frac{\Delta}{2} - \frac{\eta(\mathbf{u})}{\Delta} \right) + \pi_2 p_2(\mathbf{u}) \Phi \left( -\frac{\Delta}{2} + \frac{\eta(\mathbf{u})}{\Delta} \right) \right],$$

where $\eta(\mathbf{u}) = \log\left(\frac{\pi_1 p_1(\mathbf{u})}{\pi_2 p_2(\mathbf{u})}\right)$, and $\Delta = \frac{[\mu_1(\mathbf{u}) - \mu_2(\mathbf{u})]^\top \mathbf{b}(\mathbf{u})}{\sqrt{\mathbf{b}(\mathbf{u})^\top \Sigma(\mathbf{u}) \mathbf{b}(\mathbf{u})}}$. For a given $u \in \{0,1\}^d$, denote

$$R_{\mathbf{u}}(\Delta, \eta) = \pi_1 p_1(\mathbf{u}) \Phi\left(-\frac{\Delta}{2} - \frac{\eta(\mathbf{u})}{\Delta}\right) + \pi_2 p_2(\mathbf{u}) \Phi\left(-\frac{\Delta}{2} + \frac{\eta(\mathbf{u})}{\Delta}\right).$$

By taking the partial derivation with the respect to $\Delta$, it can be shown that

$$\frac{\partial R_{\mathbf{u}}(\Delta, c)}{\partial \Delta} = -\sqrt{\pi_1 p_1(\mathbf{u}) \pi_2(\mathbf{u}) p_2(\mathbf{u})} \exp\left\{-\left(\frac{c^2(\mathbf{u})}{\Delta^2} + \frac{1}{4}\Delta^2\right)/2\right\} < 0,$$

which implies that to minimize the expected misclassification rate, it is equivalent to look for $\mathbf{b}(\mathbf{u})$ that maximizers $\Delta(\mathbf{u})$. Now let's consider the set of zero-intercept linear classifiers defined as in (7) with $b_0(\mathbf{u}) = 0$. Consequently (9) can be written as $R(1|2) = R(2|1) = \Phi\left(-\frac{\Delta(\mathbf{u})}{2}\right)$, and the expected misclassification rate (8) reduces to:

$$R(\mathbf{b}(\mathbf{u})) = \sum_{\mathbf{u} \in \{0,1\}^d} [\pi_1 p_1(\mathbf{u}) + \pi_2 p_2(\mathbf{u})] \Phi\left(-\frac{\Delta(\mathbf{u})}{2}\right) = \Phi\left(-\frac{\Delta(\mathbf{u})}{2}\right),$$

which is also minimized when $\Delta(\mathbf{u})$ is maximized. Consequently the optimal zero-intercept classification direction is also the minimizer of (14).

On the other hand, by the definition of $\eta(\mathbf{u})$, we have:

$$\eta(\mathbf{u}) = \log \frac{\pi_1 p_1(\mathbf{u})}{\pi_2 p_2(\mathbf{u})} = \log \frac{P(\mathbf{U} = \mathbf{u}, L = 1)}{P(\mathbf{U} = \mathbf{u}, L = 2)} = \log \frac{P(L = 1|\mathbf{U} = \mathbf{u})}{P(L = 2|\mathbf{U} = \mathbf{u})}.$$

This proves the claim on $\eta(\mathbf{u})$. $\qquad\square$

## A.4 Proof of Theorem 3.1

Before we proceed to the proofs for the main theorems, we introduce some technical lemmas and establish concentration inequalities for the weighted estimators of the mean and covariance matrix functions.

**Lemma A.1.** *Let* $\mathbf{N} = (N_1, \ldots, N_d)^\top \in \{0,1\}^d$ *be a random vector with mean* $(p_1, \ldots, p_d)^\top$. *Denote* $W_i = N_i - p_i$, *and assume that there exist constants* $C > 0$ *and* $0 \le \alpha < \frac{1}{2}$ *such that* $E(\sum_{i=1}^d W_i)^k \le Ck!d^{1+\alpha(k-2)}$ *for any* $k \ge 2$. *For any* $\epsilon > 0$ *such that* $d^\alpha \epsilon \to 0$, *we have, when* $d$ *is large enough,*

$$P\left(\frac{1}{d}\left|\sum_{i=1}^d (N_i - p_i)\right| \ge \epsilon\right) \le 2\exp\{-C_1 d\epsilon^2\}. \tag{15}$$

*Proof.* Note that for any $t \leq cd^{1-\alpha}$ with some constant $c < 1$, we have,

$$
\begin{aligned}
E \exp \left\{ \frac{t}{d} \sum_{i=1}^{d} W_i \right\} &= 1 + \frac{t^2}{2! d^2} E \Big( \sum_{i=1}^{d} W_i \Big)^2 + \frac{t^3}{3! d^3} E \Big( \sum_{i=1}^{d} W_i \Big)^3 + \cdots \\
&\leq 1 + \frac{Ct^2}{d} + \frac{Ct^3}{d^{2-\alpha}} + \cdots \\
&\leq 1 + \frac{Ct^2}{d(1-c)} \\
&\leq \exp \left\{ \frac{Ct^2}{d(1-c)} \right\}.
\end{aligned}
$$

Consequently, by the general form of the Chebyshev-Markov inequalities (c.f. 6.1.a of Lin and Bai (2011)), we have, for any $\epsilon > 0$ such that $d^{\alpha} \epsilon \to 0$,

$$
P \left( \frac{1}{d} \sum_{i=1}^{d} W_i \geq \epsilon \right) \leq E \exp \left\{ \frac{t}{d} \sum_{i=1}^{d} W_i - t\epsilon \right\} \leq \exp \left\{ \frac{Ct^2}{d(1-c)} - t\epsilon \right\}.
$$

Notice that the last term in the above inequality is minimized at $t_0 = (2C)^{-1} d(1-c)\epsilon$. On the other hand, when $t = t_0$ we have $t = o(d^{1-\alpha})$. Consequently, when $d$ is large enough such that $t \leq cd^{1-\alpha}$, we have $P \left( \frac{1}{d} \sum_{i=1}^{d} W_i \geq \epsilon \right) \leq \exp \left\{ -\frac{d(1-c)\epsilon^2}{4C} \right\}$. The lemma is proved by setting $C_1 = \frac{1-c}{4C}$. $\qquad \square$

We say $W_1, \ldots, W_d$ are $m$-dependent if for any $1 \leq i \leq d$, $W_i$ is dependent to at most $m$ variables in $\{W_j : j \neq i, 1 \leq j \leq d\}$. When $W_1, \ldots, W_d$ are $m$-dependent for a constant $m$, we have $E(\sum_{i=1}^{d} W_i)^k \leq d(m+1)^{k-1}$. Hence Lemma A.1 holds for $m$-dependent sequences with $m = o(d^{\alpha})$.

**Lemma A.2.** *Under Conditions (C1) and (C2) we have, when $d$ is large enough, for any constant $C_2 > B$ and $\mathbf{u} \in \{0,1\}^d$, there exists a constant $C_1 > 0$ such that,*

$$
E \exp\{-(hd)^{-1} N_{\mathbf{u}}\} \geq \exp\{-(dh)^{-1} E N_{\mathbf{u}}\}; \tag{16}
$$

$$
E \exp\{-(hd)^{-1} N_{\mathbf{u}}\} < 2(n+d)^{-C_2} + \exp \left\{ -(dh)^{-1} E N_{\mathbf{u}} + h^{-1} \sqrt{\frac{C_1 \log(d+n)}{d}} \right\} \tag{17}
$$

$$
= \exp\{-(dh)^{-1} E N_{\mathbf{u}}\} \left( 1 + O \left( \sqrt{\frac{\log(d+n)}{h^2 d}} \right) \right).
$$

*where $B \in [0.5, 1)$ is defined as in Condition 1.*

*Proof.* (16) is a direct result of Jensen's inequality. (17) can be obtained from Lemma A.1 with $\epsilon = \sqrt{\frac{C_1 \log(d+n)}{d}}$. $\qquad \square$

Similar to Lemma 1 and Corollary 1 in Hong and Linton (2016), Lemma A.2 above provides an analogue of Bochner's Lemma for infinite-dimensional discrete variables. However, the convergence will only be obtained when $\frac{\log(d+n)}{h^2 d} \to 0$, which in return indicates that $h$ should not converge to zero with a rate faster than $O\left(\sqrt{\frac{\log(d+n)}{d}}\right)$.

**Lemma A.3.** *Under Conditions (C1)-(C3), we have, when $n_1$ and $n_2$ are large enough, for any $\mathbf{v} \in \{0,1\}^d$ and a bounded radius $r$, and any small enough $\epsilon_n > 0$, there exist constants $C_1 > 0$ and $C_2 > 0$ such that*

$$P\left(\sup_{\mathbf{u} \in B_{\mathbf{v}}(r)} \left| \frac{1}{n_1} \sum_{j=1}^{n_1} \frac{\exp\{-(dh_x)^{-1}|\mathbf{U}_j - \mathbf{u}|_1\}}{E\exp\{-(dh_x)^{-1}|\mathbf{U}_1 - \mathbf{u}|_1\}} - 1 \right| \geq \epsilon_n \right) \leq C_1 d^r \exp\left\{-C_2 n_1 \epsilon_n^2\right\}.$$

*and*

$$P\left(\sup_{\mathbf{u} \in B_{\mathbf{v}}(r)} \left| \frac{1}{n_2} \sum_{j=1}^{n_2} \frac{\exp\{-(dh_y)^{-1}|\mathbf{V}_j - \mathbf{u}|_1\}}{E\exp\{-(dh_y)^{-1}|\mathbf{V}_1 - \mathbf{u}|_1\}} - 1 \right| \geq \epsilon_n \right) \leq C_1 d^r \exp\left\{-C_2 n_2 \epsilon_n^2\right\}.$$

*Proof.* Denote $W_j(\mathbf{u}) := n_1^{-1}\left(\frac{\exp\{-(dh_x)^{-1}|\mathbf{U}_j - \mathbf{u}|_1\}}{E\exp\{-(dh_x)^{-1}|\mathbf{U}_1 - \mathbf{u}|_1\}} - 1\right)$. From Condition 2 and Lemma A.2, we have, $EW_j(\mathbf{u})^2 \leq n_1^{-2}C$ for some constant $C > 0$. By Doob's submartingale inequality we have, for any $\mathbf{u} \in \{0,1\}^d$, and any $t > 0$ such that $n_1^{-1}t$ is small enough,

$$P\left( \left| \frac{1}{n_1} \sum_{j=1}^{n_1} \frac{\exp\{-(dh_x)^{-1}|\mathbf{U}_j - \mathbf{u}|_1\}}{E\exp\{-(dh_x)^{-1}|\mathbf{U}_1 - \mathbf{u}|_1\}} - 1 \right| > \epsilon_n \right) \tag{18}$$

$$\leq \quad 2\exp\{-t\epsilon_n\}\Pi_{j=1}^{n_1}E\exp\{tW_j(\mathbf{u})\}$$

$$\leq \quad 2\exp\{-t\epsilon_n\}\Pi_{j=1}^{n_1}\{1 + t^2 EW_j(\mathbf{u})^2\}$$

$$\leq \quad 2\exp\left\{-t\epsilon_n + \sum_{j=1}^{n_1} t^2 EW_j(\mathbf{u})^2\right\}$$

$$\leq \quad 2\exp\left\{-t\epsilon_n + \frac{t^2 C}{n_1}\right\}. \tag{19}$$

Here in the second inequality we have used the fact that $EW_j(\mathbf{u}) = 0$ and $e^x \leq 1 + x + x^2$ when $x > 0$ is small enough. By setting $t = (2C)^{-1}n_1\epsilon_n$, we have:

$$P\left( \left| \frac{1}{n_1} \sum_{j=1}^{n_1} \frac{\exp\{-(dh_x)^{-1}|\mathbf{U}_j - \mathbf{u}|_1\}}{E\exp\{-(dh_x)^{-1}|\mathbf{U}_1 - \mathbf{u}|_1\}} - 1 \right| > \epsilon_n \right) \leq 2\exp\left\{-\frac{n_1\epsilon_n^2}{4C}\right\}.$$

By noticing that the cardinality of $B_{\mathbf{v}}(r)$ is less than $d^r/r!$, we have

$$P\left(\sup_{\mathbf{u} \in B_{\mathbf{v}}(r)} \left| \frac{1}{n_1} \sum_{j=1}^{n_1} \frac{\exp\{-(dh_x)^{-1}|\mathbf{U}_j - \mathbf{u}|_1\}}{E\exp\{-(dh_x)^{-1}|\mathbf{U}_1 - \mathbf{u}|_1\}} - 1 \right| \geq \epsilon_n \right) \leq \frac{2d^r}{r!}\exp\left\{-\frac{n_1\epsilon_n^2}{4C}\right\}.$$

This proves the first statement of the lemma. The second statement can be obtained similarly. $\qquad \square$

**Proof of Theorem 3.1**

*Proof.* Denote

$$W_{ji}(\mathbf{u}) := n_1^{-1}\left[ \frac{\exp\{-(dh_x)^{-1}|\mathbf{U}_j - \mathbf{u}|_1\}X_{ji}}{E\exp\{-(dh_x)^{-1}|\mathbf{U}_1 - \mathbf{u}|_1\}} - \mu_{1i}(\mathbf{u}) \right], \quad \text{and}$$

$$B_i(\mathbf{u}) := \frac{E \exp\{-(dh_x)^{-1}|\mathbf{U}_1 - \mathbf{u}|_1\} X_{1i}(\mathbf{u})}{E \exp\{-(dh_x)^{-1}|\mathbf{U}_1 - \mathbf{u}|_1\}} - \mu_{1i}(\mathbf{u}).$$

Similar to (18), for any $0 < t < T$, where $T$ is a small enough constant, we have,

$$P \left( \left| \frac{1}{n_1} \sum_{j=1}^{n_1} \frac{\exp\{-(dh_x)^{-1}|\mathbf{U}_j - \mathbf{u}|_1\} X_{ji}}{E \exp\{-(dh_x)^{-1}|\mathbf{U}_1 - \mathbf{u}|_1\}} - \mu_{1i}(\mathbf{u}) \right| > \epsilon_n \right)$$
$$\leq 2 \exp \left\{ -t\epsilon_n + tB_i(\mathbf{u}) + t^2 \sum_{j=1}^{n_1} E(W_{ji}(\mathbf{u}))^2 \right\}.$$

From Condition (C3) we have $B_i(\mathbf{u}) = \kappa_{\mathbf{u}}(h_x)$. On the other hand, by Lemma A.2 and Condition C4, we have,

$$
\begin{aligned}
E(n_1 W_{ji}(\mathbf{u}))^2 &\leq 2E \left| \frac{\exp\{-(dh_x)^{-1}|\mathbf{U}_j - \mathbf{u}|_1\} X_{ji}}{E \exp\{-(dh_x)^{-1}|\mathbf{U}_1 - \mathbf{u}|_1\}} - \frac{E \exp\{-(dh_x)^{-1}|\mathbf{U}_1 - \mathbf{u}|_1\} \mu_{1i}(\mathbf{U}_1)}{E \exp\{-(dh_x)^{-1}|\mathbf{U}_1 - \mathbf{u}|_1\}} \right|^2 \\
&\quad + 2 \left| \frac{E \exp\{-(dh_x)^{-1}|\mathbf{U}_1 - \mathbf{u}|_1\} \mu_{1i}(\mathbf{U}_1)}{E \exp\{-(dh_x)^{-1}|\mathbf{U}_1 - \mathbf{u}|_1\}} - \mu_{1i}(\mathbf{u}) \right|^2 \\
&\leq C,
\end{aligned}
$$

for some large enough constant $C > 0$. Consequently, we have

$$P \left( \left| \frac{1}{n_1} \sum_{j=1}^{n_1} \frac{\exp\{-(dh_x)^{-1}|\mathbf{U}_j - \mathbf{u}|_1\} X_{ji}}{E \exp\{-(dh_x)^{-1}|\mathbf{U}_1 - \mathbf{u}|_1\}} - \mu_{1i}(\mathbf{u}) \right| > \epsilon_n \right)$$
$$\leq 2 \exp \left\{ -t\epsilon_n + t\kappa(h_x) + \frac{Ct^2}{n_1} \right\}$$
$$\leq 2 \exp \left\{ -\frac{n_1(\epsilon_n - \kappa(h_x))^2}{4C} \right\}.$$

Together with Lemma A.3, we have

$$P \left( \sup_{1 \leq i \leq p, \mathbf{u} \in B_{\mathbf{v}}(r)} |\hat{\mu}_{1i}(\mathbf{u}) - \mu_{1i}(\mathbf{u})| > \epsilon_n \right) \leq C_1 p d^r \exp \left\{ -C_2 n_1 (\epsilon_n - \kappa(h_x))^2 \right\}.$$

This proves the first statement of the theorem. The second statement can be proved similarly. $\square$

The proof for Theorem 3.2 is similar to the proof of Theorem 3.1 provided above, and is hence omitted.

## A.5 Proof of Proposition 3.3

*Proof.* Given the true support $\mathcal{S}$, we consider the estimation

$$
\begin{aligned}
\hat{\mathbf{b}}^0(\mathbf{u}) &= \underset{\mathbf{b} \in R^q, \, \mathbf{b}_{\mathcal{S}^c} = 0}{\arg\min} \frac{1}{2} \mathbf{b}^\top \hat{\mathbf{\Omega}}(\mathbf{u}) \mathbf{b} - \hat{\mathbf{a}}(\mathbf{u})^\top \mathbf{b} + \lambda |\mathbf{b}|_1 \\
&= \underset{\mathbf{b} \in R^q, \, \mathbf{b}_{\mathcal{S}^c} = 0}{\arg\min} \frac{1}{2} \mathbf{b}_{\mathcal{S}}^\top \hat{\mathbf{\Omega}}_{\mathcal{S},\mathcal{S}}(\mathbf{u}) \mathbf{b}_{\mathcal{S}} - \hat{\mathbf{a}}_{\mathcal{S}}(\mathbf{u})^\top \mathbf{b}_{\mathcal{S}} + \lambda |\mathbf{b}_{\mathcal{S}}|_1.
\end{aligned}
$$

By the Karush-Kuhn-Tucker (KKT) condition, we have

$$\hat{\boldsymbol{\Omega}}_{\mathcal{S},\mathcal{S}}(\mathbf{u})\hat{\mathbf{b}}^0_{\mathcal{S}}(\mathbf{u}) - \hat{\mathbf{a}}_{\mathcal{S}}(\mathbf{u}) = -\lambda\mathbf{Z}, \tag{20}$$

where $\mathbf{Z}$ is the sub-gradient of $|\mathbf{b}_{\mathcal{S}}|_1$. By the definition of $\mathbf{b}^*(\mathbf{u}) := \boldsymbol{\Omega}(\mathbf{u})^{-1}\mathbf{a}(\mathbf{u})$, we have

$$\begin{pmatrix} \mathbf{a}_{\mathcal{S}}(\mathbf{u}) \\ \mathbf{a}_{\mathcal{S}^c}(\mathbf{u}) \end{pmatrix} = \begin{pmatrix} \boldsymbol{\Omega}_{\mathcal{S},\mathcal{S}}(\mathbf{u}) & \boldsymbol{\Omega}_{\mathcal{S},\mathcal{S}^c}(\mathbf{u}) \\ \boldsymbol{\Omega}_{\mathcal{S}^c,\mathcal{S}}(\mathbf{u}) & \boldsymbol{\Omega}_{\mathcal{S}^c,\mathcal{S}^c}(\mathbf{u}) \end{pmatrix} \begin{pmatrix} \mathbf{b}^*_{\mathcal{S}}(\mathbf{u}) \\ \mathbf{0} \end{pmatrix} = \begin{pmatrix} \boldsymbol{\Omega}_{\mathcal{S},\mathcal{S}}(\mathbf{u})\mathbf{b}^*_{\mathcal{S}}(\mathbf{u}) \\ \boldsymbol{\Omega}_{\mathcal{S}^c,\mathcal{S}}(\mathbf{u})\mathbf{b}^*_{\mathcal{S}}(\mathbf{u}) \end{pmatrix},$$

and hence we have $\hat{\boldsymbol{\Omega}}_{\mathcal{S},\mathcal{S}}(\mathbf{u})\hat{\mathbf{b}}^0_{\mathcal{S}}(\mathbf{u}) - \boldsymbol{\Omega}_{\mathcal{S},\mathcal{S}}(\mathbf{u})\mathbf{b}^*_{\mathcal{S}}(\mathbf{u}) + \mathbf{a}_{\mathcal{S}}(\mathbf{u}) - \hat{\mathbf{a}}_{\mathcal{S}}(\mathbf{u}) = -\lambda\mathbf{Z}$. Consequently, we have,

$$\begin{aligned} &\hat{\mathbf{b}}^0_{\mathcal{S}}(\mathbf{u}) - \mathbf{b}^*_{\mathcal{S}}(\mathbf{u}) \\ =\ & -\boldsymbol{\Omega}_{\mathcal{S},\mathcal{S}}(\mathbf{u})^{-1}\left\{\lambda\mathbf{Z} + [\hat{\boldsymbol{\Omega}}_{\mathcal{S},\mathcal{S}}(\mathbf{u}) - \boldsymbol{\Omega}_{\mathcal{S},\mathcal{S}}(\mathbf{u})]\hat{\mathbf{b}}^0_{\mathcal{S}}(\mathbf{u}) + (\mathbf{a}_{\mathcal{S}}(\mathbf{u}) - \hat{\mathbf{a}}_{\mathcal{S}}(\mathbf{u}))\right\}. \end{aligned} \tag{21}$$

By the triangle inequality, we have

$$\begin{aligned} &\|\hat{\mathbf{b}}^0_{\mathcal{S}}(\mathbf{u}) - \mathbf{b}^*_{\mathcal{S}}(\mathbf{u})\|_\infty \\ \leq\ & \|\boldsymbol{\Omega}_{\mathcal{S},\mathcal{S}}(\mathbf{u})^{-1}\|_L\Big\{\lambda\|\mathbf{Z}\|_\infty + \|[\hat{\boldsymbol{\Omega}}_{\mathcal{S},\mathcal{S}}(\mathbf{u}) - \boldsymbol{\Omega}_{\mathcal{S},\mathcal{S}}(\mathbf{u})](\hat{\mathbf{b}}^0_{\mathcal{S}}(\mathbf{u}) - \mathbf{b}^*_{\mathcal{S}}(\mathbf{u}))\|_\infty \\ & +\|[\hat{\boldsymbol{\Omega}}_{\mathcal{S},\mathcal{S}}(\mathbf{u}) - \boldsymbol{\Omega}_{\mathcal{S},\mathcal{S}}(\mathbf{u})]\mathbf{b}^*_{\mathcal{S}}(\mathbf{u}) + \mathbf{a}_{\mathcal{S}}(\mathbf{u}) - \hat{\mathbf{a}}_{\mathcal{S}}(\mathbf{u})\|_\infty\Big\} \\ \leq\ & \|\boldsymbol{\Omega}_{\mathcal{S},\mathcal{S}}(\mathbf{u})^{-1}\|_L\Big\{\lambda + \|\mathbf{b}^*(\mathbf{u})\|_0\|\hat{\boldsymbol{\Omega}}(\mathbf{u}) - \boldsymbol{\Omega}(\mathbf{u})\|_\infty\|\hat{\mathbf{b}}^0_{\mathcal{S}}(\mathbf{u}) - \mathbf{b}^*_{\mathcal{S}}(\mathbf{u})\|_\infty \\ & +\|(\hat{\boldsymbol{\Omega}}(\mathbf{u}) - \boldsymbol{\Omega}(\mathbf{u}))\mathbf{b}^*(\mathbf{u}) + \mathbf{a}(\mathbf{u}) - \hat{\mathbf{a}}(\mathbf{u})\|_\infty\Big\}, \end{aligned}$$

which implies that

$$\begin{aligned} &\|\hat{\mathbf{b}}^0_{\mathcal{S}}(\mathbf{u}) - \mathbf{b}^*_{\mathcal{S}}(\mathbf{u})\|_\infty \\ \leq\ & [1 - \|\mathbf{b}^*(u)\|_0\|\boldsymbol{\Omega}_{\mathcal{S},\mathcal{S}}(\mathbf{u})^{-1}\|_L\|\hat{\boldsymbol{\Omega}}(\mathbf{u}) - \boldsymbol{\Omega}(\mathbf{u})\|_\infty]^{-1}\|\boldsymbol{\Omega}_{\mathcal{S},\mathcal{S}}(\mathbf{u})^{-1}\|_L(\lambda + \epsilon_\mathbf{u}) \\ \leq\ & 2[1 - \|\mathbf{b}^*(u)\|_0\|\boldsymbol{\Omega}_{\mathcal{S},\mathcal{S}}(\mathbf{u})^{-1}\|_L\|\hat{\boldsymbol{\Omega}}(\mathbf{u}) - \boldsymbol{\Omega}(\mathbf{u})\|_\infty]^{-1}\|\boldsymbol{\Omega}_{\mathcal{S},\mathcal{S}}(\mathbf{u})^{-1}\|_L\lambda, \end{aligned} \tag{22}$$

where in the last inequality we have used the fact (from the conditions) that $\epsilon_\mathbf{u} < \lambda$. Next, we complete the proof by showing that $\hat{\mathbf{b}}^0(\mathbf{u})$ is exactly the minimizer of (10). By the KKT condition, it is sufficient to show

$$\|(\hat{\boldsymbol{\Omega}}(\mathbf{u})\hat{\mathbf{b}}^0(\mathbf{u}) - \hat{\mathbf{a}}(\mathbf{u}))_{\mathcal{S}}\|_\infty \leq \lambda, \text{ and} \tag{23}$$

$$\|(\hat{\boldsymbol{\Omega}}(\mathbf{u})\hat{\mathbf{b}}^0(\mathbf{u}) - \hat{\mathbf{a}}(\mathbf{u}))_{\mathcal{S}^c}\|_\infty < \lambda. \tag{24}$$

(23) is a direct result of (20). For (24), we have

$$(\hat{\boldsymbol{\Omega}}(\mathbf{u})\hat{\mathbf{b}}^0(\mathbf{u}) - \hat{\mathbf{a}}(\mathbf{u}))_{\mathcal{S}^c}$$
$$= \hat{\boldsymbol{\Omega}}_{\mathcal{S}^c,\mathcal{S}}(\mathbf{u})\hat{\mathbf{b}}^0_{\mathcal{S}}(\mathbf{u}) - \hat{\mathbf{a}}_{\mathcal{S}^c}(\mathbf{u})$$
$$= \hat{\boldsymbol{\Omega}}_{\mathcal{S}^c,\mathcal{S}}(\mathbf{u})\hat{\mathbf{b}}^0_{\mathcal{S}}(\mathbf{u}) - \boldsymbol{\Omega}_{\mathcal{S}^c,\mathcal{S}}(\mathbf{u})\mathbf{b}^*_{\mathcal{S}}(\mathbf{u}) + \mathbf{a}_{\mathcal{S}^c}(\mathbf{u}) - \hat{\mathbf{a}}_{\mathcal{S}^c}(\mathbf{u})$$
$$= \hat{\boldsymbol{\Omega}}_{\mathcal{S}^c,\mathcal{S}}(\mathbf{u})(\hat{\mathbf{b}}^0_{\mathcal{S}}(\mathbf{u}) - \mathbf{b}^*_{\mathcal{S}}(\mathbf{u})) + [\hat{\boldsymbol{\Omega}}_{\mathcal{S}^c,\mathcal{S}}(\mathbf{u}) - \boldsymbol{\Omega}_{\mathcal{S}^c,\mathcal{S}}(\mathbf{u})]\mathbf{b}^*_{\mathcal{S}}(\mathbf{u}) + \mathbf{a}_{\mathcal{S}^c}(\mathbf{u}) - \hat{\mathbf{a}}_{\mathcal{S}^c}(\mathbf{u})$$
$$= [\hat{\boldsymbol{\Omega}}_{\mathcal{S}^c,\mathcal{S}}(\mathbf{u}) - \boldsymbol{\Omega}_{\mathcal{S}^c,\mathcal{S}}(\mathbf{u})](\hat{\mathbf{b}}^0_{\mathcal{S}}(\mathbf{u}) - \mathbf{b}^*_{\mathcal{S}}(\mathbf{u})) + \boldsymbol{\Omega}_{\mathcal{S}^c,\mathcal{S}}(\mathbf{u})\boldsymbol{\Omega}_{\mathcal{S},\mathcal{S}}(\mathbf{u})^{-1}\{\boldsymbol{\Omega}_{\mathcal{S},\mathcal{S}}(\mathbf{u})(\hat{\mathbf{b}}^0_{\mathcal{S}}(\mathbf{u}) - \mathbf{b}^*_{\mathcal{S}}(\mathbf{u}))\}$$
$$\quad + \{(\hat{\boldsymbol{\Omega}}(\mathbf{u}) - \boldsymbol{\Omega}(\mathbf{u}))\mathbf{b}^*(\mathbf{u}) + \mathbf{a}(\mathbf{u}) - \hat{\mathbf{a}}(\mathbf{u})\}_{\mathcal{S}^c}.$$

Together with (21) and (22), we have

$$\|(\hat{\boldsymbol{\Omega}}(\mathbf{u})\hat{\mathbf{b}}^0(\mathbf{u}) - \hat{\mathbf{a}}(\mathbf{u}))_{\mathcal{S}^c}\|_\infty$$
$$\leq \|\mathbf{b}^*(\mathbf{u})\|_0\|\hat{\boldsymbol{\Omega}}(\mathbf{u}) - \boldsymbol{\Omega}(\mathbf{u})\|_\infty\|\hat{\mathbf{b}}^0_{\mathcal{S}}(\mathbf{u}) - \mathbf{b}^*_{\mathcal{S}}(\mathbf{u})\|_\infty$$
$$\quad + \|\boldsymbol{\Omega}_{\mathcal{S}^c,\mathcal{S}}(\mathbf{u})\boldsymbol{\Omega}_{\mathcal{S},\mathcal{S}}(\mathbf{u})^{-1}\|_L(\lambda + \epsilon_{\mathbf{u}} + \|\mathbf{b}^*(\mathbf{u})\|_0\|\hat{\boldsymbol{\Omega}}(\mathbf{u}) - \boldsymbol{\Omega}(\mathbf{u})\|_\infty\|\hat{\mathbf{b}}^0_{\mathcal{S}}(\mathbf{u}) - \mathbf{b}^*_{\mathcal{S}}(\mathbf{u})\|_\infty) + \epsilon_{\mathbf{u}}$$
$$\leq \frac{(1 + \|\boldsymbol{\Omega}_{\mathcal{S}^c,\mathcal{S}}(\mathbf{u}))\boldsymbol{\Omega}_{\mathcal{S},\mathcal{S}}(\mathbf{u}))^{-1}\|_L)(\lambda + \epsilon_{\mathbf{u}})}{1 - \|\mathbf{b}^*(\mathbf{u}))\|_0\|\boldsymbol{\Omega}_{\mathcal{S},\mathcal{S}}(\mathbf{u}))^{-1}\|_L\|\hat{\boldsymbol{\Omega}}(\mathbf{u})) - \boldsymbol{\Omega}(\mathbf{u}))\|_\infty} - \lambda$$
$$= \left\{ \epsilon_{\mathbf{u}} - \frac{1 - \|\boldsymbol{\Omega}_{\mathcal{S}^c,\mathcal{S}}(\mathbf{u}))\boldsymbol{\Omega}_{\mathcal{S},\mathcal{S}}(\mathbf{u}))^{-1}\|_L - 2\|\mathbf{b}^*(\mathbf{u}))\|_0\|\boldsymbol{\Omega}_{\mathcal{S},\mathcal{S}}(\mathbf{u}))^{-1}\|_L\|\hat{\boldsymbol{\Omega}}(\mathbf{u})) - \boldsymbol{\Omega}(\mathbf{u}))\|_\infty}{1 + |\boldsymbol{\Omega}_{\mathcal{S}^c,\mathcal{S}}(\mathbf{u}))\boldsymbol{\Omega}_{\mathcal{S},\mathcal{S}}(\mathbf{u}))^{-1}\|_L}\lambda \right\}$$
$$\quad \times \left\{ \frac{1 + \|\boldsymbol{\Omega}_{\mathcal{S}^c,\mathcal{S}}(\mathbf{u})\boldsymbol{\Omega}_{\mathcal{S},\mathcal{S}}(\mathbf{u})^{-1}\|_L}{1 - \|\mathbf{b}^{*(\mathbf{u})}\|_0\|\boldsymbol{\Omega}_{\mathcal{S},\mathcal{S}}(\mathbf{u})^{-1}\|_L\|\hat{\boldsymbol{\Omega}}(\mathbf{u}) - \boldsymbol{\Omega}(\mathbf{u})\|_\infty} \right\} + \lambda.$$

Under the Condition that

$$2[1 - \|\boldsymbol{\Omega}_{\mathcal{S}^c,\mathcal{S}}(\mathbf{u})\boldsymbol{\Omega}_{\mathcal{S},\mathcal{S}}(\mathbf{u})^{-1}\|_L - 2\|\mathbf{b}^*(\mathbf{u})\|_0\|\boldsymbol{\Omega}_{\mathcal{S},\mathcal{S}}(\mathbf{u})^{-1}\|_L\|\hat{\boldsymbol{\Omega}}(\mathbf{u}) - \boldsymbol{\Omega}(\mathbf{u})\|_\infty]^{-1}\epsilon_{\mathbf{u}} < \lambda,$$

we have $\|(\hat{\boldsymbol{\Omega}}(\mathbf{u})\hat{\mathbf{b}}^0(\mathbf{u}) - \hat{\mathbf{a}}(\mathbf{u}))_{\mathcal{S}^c}\|_\infty < \lambda$. Consequently, $\hat{\mathbf{b}}(\mathbf{u}) = \hat{\mathbf{b}}^0(\mathbf{u})$. Lastly, note that the inequality conditions in this proposition hold for all $\mathbf{u} \in B_{\mathbf{v}}(r)$, we conclude that the model selection consistency and the bound (22) hold for all $\mathbf{u} \in B_{\mathbf{v}}(r)$. □

## A.6  Proof of Theorem 3.4

*Proof.* Set $\epsilon_n = C\sqrt{\frac{\log(p+d)}{n}} + \kappa(h)$ for some large enough constant $C > 0$. From Theorems 3.1 and 3.2 we have when $C$ is large enough, with probability larger than $1 - O((p+d)^{-1})$, $\sup_{\mathbf{u} \in B_{\mathbf{v}}(r)} \|\hat{\mu}_1(\mathbf{u}) - \mu_1(\mathbf{u})\|_\infty = O(\epsilon_n)$ and $\|\hat{\Sigma}(\mathbf{u}) - \Sigma(\mathbf{u})\|_\infty = O(\epsilon_n)$. Consequently, when $C >$ is large enough and $\epsilon_n \to 0$, we have

$$\sup_{\mathbf{u} \in B_{\mathbf{v}}(r)} \{\|\Sigma_{\mathcal{S}^c,\mathcal{S}}(\mathbf{u})\Sigma_{\mathcal{S},\mathcal{S}}(\mathbf{u})^{-1}\|_L + 2\|\beta(\mathbf{u})\|_0\|\Sigma_{\mathcal{S},\mathcal{S}}(\mathbf{u})^{-1}\|_L\|\hat{\Sigma}(\mathbf{u}) - \Sigma(\mathbf{u})\|_\infty\} < 1,$$

and

$$\sup_{\mathbf{u} \in B_{\mathbf{v}}(r)} 2[1 - \|\Sigma_{\mathcal{S}^c,\mathcal{S}}(\mathbf{u})\Sigma_{\mathcal{S},\mathcal{S}}(\mathbf{u})^{-1}\|_L - 2\|\beta(\mathbf{u})\|_0\|\Sigma_{\mathcal{S},\mathcal{S}}(\mathbf{u})^{-1}\|_L\|\hat{\Sigma}(\mathbf{u}) - \Sigma(\mathbf{u})\|_\infty]^{-1}\epsilon_{\mathbf{u}}$$
$$< \lambda_\beta.$$

The theorem then follows from Proposition 3.3. □

## A.7  Proof of Theorem 3.5

*Proof.* Denote the objective function on the right hand side of (5) as $L_n(\eta)$. The true linear function $\eta(\mathbf{W}) = A_0 + \mathbf{A}^\top \mathbf{W}$ is the minimizer of the expected risk:

$$El(\tilde{\eta}; \mathbf{W}, L)) = E\{-(2-L)\tilde{\eta}(\mathbf{W}) + \log(1 + \exp\{\tilde{\eta}(\mathbf{W})\})\}.$$

Let $\hat{\mathcal{M}} := \{j : \hat{A}_j \neq 0\}$. Using similar arguments as in the proof of Proposition 3.3, we can first show that under Conditions (C5) and (C6), $P(\hat{\mathcal{M}} = \mathcal{M}) \to 1$. For convenience we shall assume hereafter in this proof that $\hat{\mathcal{M}} = \mathcal{M}$.

Let $\mathcal{B}_\eta(\delta)$ be the set of linear functions $\bar{\eta}(\mathbf{w}) = \bar{A}_0 + \bar{\mathbf{A}}_1^\top \mathbf{w}$ , such that $\sum_{i=0}^d |\bar{A}_i - A_i| = \delta$ for some $\delta > 0$. Here $\bar{A}_i$ is the $i$-th element of $\bar{\mathbf{A}}_1$, and $\bar{A}_i = 0$ for $i \in \mathcal{M}^c$. With some abuse of notations, let $W_i = \mathbf{U}_i, \tilde{W}_i = \binom{1}{W_i}$ for $i = 1\ldots, n_1$, $W_{n_1+i} = \mathbf{V}_i, \tilde{W}_{n_1+i} = \binom{1}{W_{n_1+i}}$ for $i = 1\ldots, n_2$, and denote $\mathbf{W}_n = (\tilde{W}_1, \ldots, \tilde{W}_n)$. The Hessian matrix of $L_n$ is then given as:

$$H_n(\eta) = \sum_{i=1}^n \tilde{W}_i \tilde{W}_i^\top \frac{\exp\{\eta(W_i)\}}{(1 + \exp\{\eta(W_i)\})^2}.$$

By Taylor expansion we have, there exists an $\eta^*(\mathbf{w}) = A_0^* + \mathbf{w}^\top \mathbf{A}_1^*$, such that $A_0^* \in [\bar{A}_0, A_0]$, $A_i^* \in [\bar{A}_i, A_i]$ for $i \in \mathcal{M}$, $A_i^* = 0$ for $i \in \mathcal{M}^c$, and

$$L_n(\bar{\eta}) = L_n(\eta) + L_n'(\eta) \begin{pmatrix} \bar{A}_0 - A_0 \\ \bar{\mathbf{A}}_1 - \mathbf{A}_1 \end{pmatrix} + \begin{pmatrix} \bar{A}_0 - A_0 \\ \bar{\mathbf{A}}_1 - \mathbf{A}_1 \end{pmatrix}^\top H_n(\eta^*) \begin{pmatrix} \bar{A}_0 - A_0 \\ \bar{\mathbf{A}}_1 - \mathbf{A}_1 \end{pmatrix}.$$

Note that from Bernstein's inequality (Lin and Bai, 2011) and the fact that $EL_n'(\eta) = \mathbf{0}$, we have, there exists a constant $C > 0$ such that with probability larger than $1 - O(d^{-1})$,

$$\left| L_n'(\eta) \begin{pmatrix} \bar{A}_0 - A_0 \\ \bar{\mathbf{A}}_1 - \mathbf{A}_1 \end{pmatrix} \right| \leq C\delta M \sqrt{\frac{\log d}{n}},$$

holds for all $\bar{\eta} \in \mathcal{B}_\eta(\delta)$. On the other hand, using Condition (C5) and the fact that $(\bar{A}_0 - A_0)^2 + \|\bar{\mathbf{A}}_1 - \mathbf{A}_1\|_2^2 \geq M^{-1}\delta^2$, we have with probability larger than $1 - O(d^{-1})$,

$$\begin{pmatrix} \bar{A}_0 - A_0 \\ \bar{\mathbf{A}}_1 - \mathbf{A}_1 \end{pmatrix}^\top H_n(\eta^*) \begin{pmatrix} \bar{A}_0 - A_0 \\ \bar{\mathbf{A}}_1 - \mathbf{A}_1 \end{pmatrix} \geq \frac{\delta^2}{Me^M},$$

holds for all $\bar{\eta} \in \mathcal{B}_\eta(\delta)$. Consequently, by choosing $\delta = C_1 M^2 e^M \sqrt{\frac{\log d}{n}}$ for some large enough constant $C_1 > 0$, we have with probability larger than $1 - O(d^{-1})$, $L_n(\bar{\eta}) < L_n(\eta)$ holds for all $\bar{\eta} \in \mathcal{B}_\eta(\delta)$. Consequently, by continuity and convexity of the objective function $L_n$, we conclude that with probability tending to 1, the minimizer $\hat{\eta}$ of $L_n$ must lie inside the $L_1$-ball with radius $\delta$, i.e., $|\hat{\eta}(\mathbf{w}) - \eta(\mathbf{w})| \leq \sum_{i=0}^d |\hat{A}_i - A_i| < \delta$. This proves the theorem. □

## A.8   Proof of Theorem 3.6

*Proof.* By definition we have:

$$\hat{D}(\mathbf{Z}, \mathbf{u}) - D(\mathbf{Z}, \mathbf{u}) = \left\{ \mathbf{Z}^\top [\hat{\beta}(\mathbf{u}) - \beta(\mathbf{u})] - \frac{1}{2}\hat{\beta}(\mathbf{u})^\top [\hat{\mu}_1(\mathbf{u}) + \hat{\mu}_2(\mathbf{u})] \right.$$

$$\left. + \frac{1}{2}\beta(\mathbf{u})^\top [\mu_1(\mathbf{u}) + \mu_2(\mathbf{u})] + [\hat{\eta}(\mathbf{u}) - \eta(\mathbf{u})] \right\}.$$

Write $\mathbf{Z} = (Z_1, \cdots, Z_p)^\top$. From Theorem 3.4, we have:

$$\sup_{\mathbf{u} \in B_\mathbf{v}(r)} |\mathbf{Z}^\top [\hat{\beta}(\mathbf{u}) - \beta(\mathbf{u})]|$$

$$\leq \sup_{\mathbf{u} \in B_\mathbf{v}(r)} \sum_{i=1}^p |Z_i [\hat{\beta}(\mathbf{u}) - \beta(\mathbf{u})]_i| = O_p\left( s_\mathbf{u} M_r \left( \sqrt{\frac{\log(p+d)}{n}} + \kappa(h) \right) \right).$$

Similarly, from Theorems 1 and 3 we have

$$\sup_{\mathbf{u} \in B_\mathbf{v}(r)} |\hat{\beta}(\mathbf{u})^\top [\hat{\mu}_1(\mathbf{u}) + \hat{\mu}_2(\mathbf{u})] - \beta(\mathbf{u})^\top [\mu_1(\mathbf{u}) + \mu_2(\mathbf{u})]|$$

$$= O_p\left( \left(s + \sup_{\mathbf{u} \in B_\mathbf{v}(r)} |\beta(\mathbf{u})|_1\right) M_r \left( \sqrt{\frac{\log(p+d)}{n}} + \kappa(h) \right) \right).$$

Together with Theorem 3.5, we have:

$$\sup_{\mathbf{u} \in B_\mathbf{v}(r)} |\hat{D}(\mathbf{Z}; \mathbf{u}) - D(\mathbf{Z}; \mathbf{u})| \qquad (25)$$

$$= O_p\left( \left(s + \sup_{\mathbf{u} \in B_\mathbf{v}(r)} |\beta(\mathbf{u})|_1\right) M_r \left( \sqrt{\frac{\log(p+d)}{n}} + \kappa(h) \right) + M^2 e^M \sqrt{\frac{\log d}{n}} \right).$$

Note that

$$\hat{R}(1|2) = P(\hat{D}(\mathbf{Z}; \mathbf{u}) > 0 | \mathbf{Z} \in N(\mu_2(\mathbf{u}), \Sigma(\mathbf{u})))$$

$$= P(D(\mathbf{Z}; \mathbf{u}) > D(\mathbf{Z}; \mathbf{u}) - \hat{D}(\mathbf{Z}; \mathbf{u}) | \mathbf{Z} \in N(\mu_2(\mathbf{u}), \Sigma(\mathbf{u}))).$$

On the other hand, for a given $\mathbf{u}$, denote the density of $D(\mathbf{Z}; \mathbf{u})$ as $F_i(\cdot; \mathbf{u})$ for $\mathbf{Z}$ in class $i = 1, 2$. Under Conditions (C5) and (C6), we have $F_i(\cdot; \mathbf{u})$ is bounded from above. Together with (25), we conclude that

$$\hat{R}(1|2) - R(1|2)$$

$$= O_p\left( \int_0^{\left(s + \sup_{\mathbf{u} \in B_\mathbf{v}(r)} |\beta(\mathbf{u})|_1\right) M_r \left( \sqrt{\frac{\log(p+d)}{n}} + \kappa(h) \right) + M^2 e^M \sqrt{\frac{\log d}{n}}} F_2(z; \mathbf{u}) dz \right)$$

$$= O_p\left( \left(s + \sup_{\mathbf{u} \in B_\mathbf{v}(r)} |\beta(\mathbf{u})|_1\right) M_r \left( \sqrt{\frac{\log(p+d)}{n}} + \kappa(h) \right) + M^2 e^M \sqrt{\frac{\log d}{n}} \right).$$

The theorem is proved by establishing a similar bound for $\hat{R}(2|1) - R(2|1)$. $\quad\square$

