# OpenReview forum: "Linear Discriminant Analysis with High-dimensional Mixed Variables"
_SLADS/Section_A — Accepted by SLADS_Section_A_

### Review · Reviewer_PHmv · 2026-05-26

**Summary Of Contributions:**

The paper proposes a semiparametric location model (SLM) for classifying high-dimensional datasets that contain both continuous and categorical variables. It models continuous variables as Gaussian conditional on discrete variables, resolving the curse of dimensionality  via kernel smoothing. The authors show that the classification direction and intercept parameters can be estimated independently via penalized likelihoods. Theoretical analyses establish concentration inequalities for kernel estimators in high-dimensional discrete spaces, yielding convergence to optimal Bayes risk.

**Audience:**

Yes

**Broader Impact Concerns:**

Given that the methodology is explicitly motivated by and applied to clinical datasets (e.g., Hepatocellular Carcinoma and Breast Cancer Gene Expression Profiles), there are risks of downstream algorithmic bias. Parsemonious approximations and uniform bandwidth selections could inadvertently smooth over rare demographic or clinical subpopulations, potentially leading to disproportionately high misclassification rates for minority groups in medical decision-making.

**Claims And Evidence:**

Yes

**Requested Changes:**

1. Can the authors further validate the common covariance assumption either theoretically or empirically?

2. Since the authors argue that the extension to non-binaray variable is straightforward, can the authors discuss more in this direction? For example, how does the multi-category impact theory? Is it only through a bigger d? Can the authors provide one concrete real data application with multiple categories?

3. How does the method extend to ordinal data? Dummy variable is not ideal for ordinal data since it breaks the order.

4. Introduction is way too long. I appreciate the clear motivation and literature review in the introduction. In particular, it's clear to me why treating the binary variable as continuous is suboptimal, and why LDA doesn't work. However, some details about the SLM can be removed or reduced, especially those on Page 4.

5. Can the authors add more state-of-the-art competitors as listed above?

**Strengths And Weaknesses:**

**Strengths**

1. The approach addresses a pervasive challenge in integrating mixed high-dimensional data, heavily motivated by practical applications in clinical diagnostics and integrative genomics.

2. Proposition 2.1 introduces an elegant theoretical separation for estimating the direction and intercept, significantly reducing the  computational complexity.

3. The extension of kernel smoothing theory to high-dimensional binary covariates using small ball probability analysis represents an interesting independent theoretical contribution.

**Weakness**

1. The assumption of a common covariance matrix for conditional Gaussian distributions across different classes is restrictive and may fail in highly heterogeneous real-world data.

2. The methodology heavily relies on binary discrete variables; while dummy variable transformations are briefly suggested, the impact of expanding multi-category variables on kernel smoothing metrics is underexplored.

3. Using a first-order linear approximation for the intercept parameter prioritizes parsimony but risks ignoring significant higher-order interactions among categorical features.

4. Some strong competing methods are missing in the experiment section, including:

Mixed-type naive Bayes

XGBoost or other boosting method;

Factor analysis of mixed data (or other embedding methods) to embed the mixed features into a continuous space followed by a standard classifier,

---

> ### Author Response · Authors · 2026-06-12
> **Response to reviewer PHmv (part 1)**
>
> We thank the reviewers for their careful reading and constructive comments.
>
> > 1.Can the authors further validate the common covariance assumption either theoretically or empirically?
>
> **Reply:**
> We thank the reviewer for raising this important point. We agree that the common covariance assumption is a modeling restriction. In the current paper, this assumption is inherited from the LDA framework. Its role is to make the conditional Bayes rule linear in the continuous variable $Z$ at each location $U=u$, leading to the discriminant direction
>
> $$
> \\beta(u)=\\Sigma^{-1}(u)\\{\\mu_1(u)-\\mu_2(u)\\}.
> $$
>
> If the two classes have substantially different conditional covariance matrices, the corresponding Bayes rule would generally be quadratic rather than linear. This would lead to a semiparametric QDA-type location model, which is beyond the scope of the present paper.
>
> We have revised Section 5 to clarify this modeling limitation. In particular, we now state that the proposed method should be viewed as a semiparametric extension of sparse LDA to high-dimensional mixed variables, and that settings with strong class-specific covariance heterogeneity would require a QDA-type extension. Such an extension is an interesting direction for future work, but it is not pursued in the current paper.
>
> > 2.Since the authors argue that the extension to non-binaray variable is straightforward, can the authors discuss more in this direction? For example, how does the multi-category impact theory? Is it only through a bigger d? Can the authors provide one concrete real data application with multiple categories?
>
> **Reply:**  We thank the reviewer for raising this point. We agree that the extension to multi-category variables should be discussed more explicitly. In the original manuscript, we briefly mentioned that variables with more than two categories can be transformed into binary variables by dummy variable encoding, but we did not sufficiently explain how this affects the theory and the real-data applications. For nominal multi-category variables, the approach used in the paper is to encode them into binary dummy variables and then apply the proposed method to the expanded binary vector. If the original categorical variables have $K_1,\\ldots,K_q$ levels, then the expanded binary dimension becomes
> $$
> d_{\\rm bin}=\\sum_{j=1}^q (K_j-1),
> $$
>  up to the precise convention used for the reference level and missing categories. Thus, the impact is partly through a larger binary dimension $d_{\\rm bin}$, but not only through the numerical value of $d$. The dummy encoding also changes the Hamming geometry used by the kernel weights, and the regularity conditions in the theory should be interpreted for the expanded binary vector. In particular, the small-ball probability and the concentration results depend on the expanded binary representation, while the sparsity condition for the intercept depends on the number of active dummy variables after encoding. We have added a discussion of this point to the manuscript. We have also revised the real-data section to make clear that several of our real-data examples already contain multi-category variables. In particular, the Cylinder Bands data contain 20 categorical variables, which were transformed into 465 binary variables via dummy variable encoding. The Heart-Disease data contain 7 categorical variables transformed into 12 binary variables. The Hepatitis data contain 12 categorical variables transformed into 22 binary variables. The Australian Credit Card Approval data contain 8 categorical attributes transformed into 28 binary variables. The German Credit data contain 13 categorical variables transformed into 41 binary variables. Therefore, the existing real-data study already includes concrete applications with multi-category categorical variables; we have revised Section 4.4 to emphasize this point. Specifically, we added the following discussion to Section 5:
> > "For nominal categorical variables with more than two levels, we use dummy variable encoding and apply the proposed method to the expanded binary vector. If the original categorical variables have $K_1,\\ldots,K_q$ levels, the resulting binary dimension is $d_{\\rm bin}=\\sum_{j=1}^q(K_j-1)$, up to the convention for reference levels and missing categories. The effect of multi-category variables is therefore not merely notational. The expanded dimension enters the Hamming distance used in the kernel weights, the integrated small-ball probability in the normalization, and the sparsity level of the logistic approximation for $\\eta(u)$. The regularity conditions in Section 3 should be interpreted as conditions on the expanded binary representation. In particular, after dummy encoding, the components of the expanded vector may have deterministic group constraints, but the concentration and smoothness requirements are imposed on the resulting binary vector used by the estimator."

---

> ### Author Response · Authors · 2026-06-12
> **Response to reviewer PHmv (part 2)**
>
> >3. How does the method extend to ordinal data? Dummy variable is not ideal for ordinal data since it breaks the order.
>
> **Reply:**
> We thank the reviewer for this helpful comment. We agree that ordinary dummy encoding is not ideal for ordinal variables, because it treats different levels as nominal categories and does not preserve their natural ordering. Since the proposed method is based on a distance over the discrete variables, ordinal variables can be incorporated by using an order-preserving binary encoding or, equivalently, an order-preserving distance.
>
> In the revised manuscript, we now distinguish nominal multi-category variables from ordinal variables. For a nominal variable with $K$ levels, ordinary dummy encoding is appropriate. For an ordinal variable with ordered levels $1,\\ldots,K$, we propose to use a cumulative binary encoding
>
> $$
> 1\\mapsto (0,\\ldots,0),\\quad
> 2\\mapsto (1,0,\\ldots,0),\\quad
> \\ldots,\\quad
> K\\mapsto (1,\\ldots,1).
> $$
>
> Under this encoding, if $c(r)$ denotes the binary vector corresponding to level $r$, then
>
> $$
> \\|c(r)-c(s)\\|_1=|r-s|.
> $$
>
> Thus, the Hamming distance on the expanded binary vector preserves the ordinal distance between levels. This allows the proposed kernel smoothing procedure to remain within the same binary-vector framework while respecting the ordering of ordinal variables.
>
> We have added this clarification to Section 5. This modification does not change the main theoretical framework: the method is still applied to an expanded binary vector, but for ordinal variables the expansion is chosen to preserve order. The regularity conditions in Section 3 should then be interpreted with respect to this order-preserving binary representation.
>
> >4. Introduction is way too long. I appreciate the clear motivation and literature review in the introduction. In particular, it's clear to me why treating the binary variable as continuous is suboptimal, and why LDA doesn't work. However, some details about the SLM can be removed or reduced, especially those on Page 4.
>
> **Reply:**
> We thank the reviewer for this helpful suggestion. We have shortened the Introduction by reducing the technical and implementation-level details of the proposed SLM, especially those originally included on Page 4. We kept the motivating example and the literature review, since these parts explain why mixed variables should not be treated as purely continuous variables and why ordinary LDA is not sufficient in the high-dimensional mixed-variable setting. However, we agree that the original Introduction contained too many details about the proposed estimator.
>
> In the revised Introduction, we retain only the essential formulas needed to make the model and the subsequent sections self-contained. Specifically, we keep the logit representation of $\\eta(u)$, the first-order approximation
>
> $$
> \\eta(u)=A_0+\\sum_{j=1}^d A_j u_j,
> $$
>
> the corresponding penalized logistic estimator for $(A_0,A)$, and the penalized quadratic estimator for the location-dependent discriminant direction $\\beta(u)$. These formulas are needed because later sections refer to them directly. At the same time, we removed or substantially reduced the longer discussion of the kernel smoothing construction, the small-ball probability argument, the analogue of Bochner's Lemma, bandwidth interpretation, and the theoretical consequences of these constructions. These details are now left to Sections 2 and 3, where they are developed more systematically.
>
> We also streamlined the contribution paragraph to avoid repeating technical details that are developed later in the paper.

---

> ### Author Response · Authors · 2026-06-12
> **Response to reviewer PHmv (part 3)**
>
> > 5.Can the authors add more state-of-the-art competitors as listed above?
>
> **Reply:**
> We thank the reviewer for this suggestion. We have added the requested additional competitors to both the simulation and real-data studies. Specifically, we added factor analysis of mixed data (FAMD), gradient boosting (GB), and mixed-type naive Bayes (MNB).
>
> The additional methods are described in Section 4.2 as follows:
>
> > "FAMD: Factor analysis of mixed data. The mixed variables are first embedded into a low-dimensional continuous representation by factor analysis of mixed data, and a standard classifier is then fitted using the extracted components."
>
> > "GB: Gradient boosting classifier. We apply a tree-based gradient boosting classifier to the combined feature matrix after the categorical variables are encoded as binary variables."
>
> > "MNB: Mixed-type naive Bayes. This classifier uses Gaussian likelihoods for the continuous variables and Bernoulli or multinomial likelihoods for the categorical variables, together with a conditional independence assumption given the class label."
>
> The updated simulation results are reported in Table 1, and the updated real-data results are reported in Table 2. These additional comparisons make the empirical study more complete. They also show that the proposed method is competitive with a broader set of mixed-data and machine-learning benchmarks, while not uniformly dominating them in every setting.

---

> > ### Comment · Reviewer_PHmv · 2026-06-28
> >
> > My previous concerns were all addressed. I'd like to thank the authors for the detailed responses and updating the manuscript accordingly.

---

### Review · Reviewer_hrWB · 2026-05-29

**Summary Of Contributions:**

This work studies linear discriminant analysis with both continuous and discrete covariates. Assuming that, conditional on the discrete covariates, the continuous covariates follow a Gaussian distribution, the Bayes optimal classification rule is expressed as a semiparametric function of the discrete covariates. Kernel methods are used to estimate the nonparametric components, and theoretical guarantees are established.

**Audience:**

Yes

**Broader Impact Concerns:**

None.

**Claims And Evidence:**

Yes

**Requested Changes:**

**Major comments:**

1. Method interpretation. While the proposed method appears natural, the use of kernel smoothing may lead to slow convergence rates, especially when the sample size is not very large. A straightforward alternative would be to fit a logistic regression using both continuous and discrete covariates. As a nonparametric alternative, k‑nearest neighbors (KNN) could also be applied. Under what scenarios should the proposed semiparametric method be preferred over these benchmarks? Please clarify.

2. Theoretical results should be further clarified. The theoretical results need further clarification. When kernel smoothing is involved, the smoothness conditions (C3) are not presented in the standard way. Could the authors state the Hölder or Lipschitz smoothness conditions on the true functions explicitly?

3. Numerical results. The interpretation of the numerical results should be more careful and reliable. Section 4.3 states: "in Table 1, from which we observe that our proposed SLM classifier outperforms other classifiers." However, under the first three configurations of Model 1 and the last three configurations of Model 2, the DSDA method outperforms the proposed SLM. Moreover, the PLG method also yields better performance in some settings. There is no clear pattern indicating when the proposed method is superior. The authors may increase the number of Monte Carlo repetitions and provide a more detailed discussion of the empirical advantages of their method.

**Minor comments:**

1. What is the difference between $\eta(U)$ in (3) and $b_0(U)$ in (7)?

2. Highlight the best results in Table 1.

3. Please provide a pseudo algorithm summary of the proposed method.

**Strengths And Weaknesses:**

**Strengths**

1. The mathematical derivations are solid and clearly presented. The theoretical results and numerical experiments are thorough.

2. The method extends classical linear discriminant analysis in a meaningful way and has clear practical applications.

**Weekness**

1. The method requires further interpretation, particularly regarding the specific scenarios in which it outperforms existing approaches.

2. The numerical results are not entirely convincing, as no clear pattern emerges to indicate when the proposed method is superior.

---

> ### Author Response · Authors · 2026-06-12
> **Response to reviewer hrWB (part 1)**
>
> We thank the reviewers for their careful reading and constructive comments.
>
> > Method interpretation. While the proposed method appears natural, the use of kernel smoothing may lead to slow convergence rates, especially when the sample size is not very large. A straightforward alternative would be to fit a logistic regression using both continuous and discrete covariates. As a nonparametric alternative, k-nearest neighbors (KNN) could also be applied. Under what scenarios should the proposed semiparametric method be preferred over these benchmarks? Please clarify.
>
> **Reply:**
> We thank the reviewer for pointing out that the interpretation of the proposed method should be made clearer. We agree that kernel smoothing can be statistically demanding when the sample size is not large, and we do not intend to claim that the proposed method should uniformly outperform logistic regression or KNN.
>
> Our motivation is closely tied to two points that were already discussed in the original manuscript. First, simply treating the categorical variables as ordinary continuous covariates may lose the mixed-variable structure. In the simple example in the Introduction, there is one continuous variable $X$ and one binary variable $U$. If $U$ is simply treated as a continuous covariate and the best linear classifier is used, the resulting misclassification rate is
>
> $$
> \\frac{1}{2}\\left\\{\\frac{1}{2}+\\Phi(-1)\\right\\},
> $$
>
> which is more than twice the Bayes misclassification rate $\\Phi(-1)$. This example illustrates that the categorical and continuous variables may need to be handled differently, and that their interaction can be essential for classification. We have revised the relevant paragraph to make this point explicit in relation to logistic regression.
>
> Second, the problem considered in the paper is high-dimensional in both the continuous component $Z$ and the discrete component $U$. As discussed in the Introduction, methods justified under fixed-dimensional asymptotics may behave poorly in high-dimensional classification problems; in particular, Bickel and Levina (2004) showed that classical discriminant rules can perform poorly when the number of variables is much larger than the number of observations. This is why the proposed method combines the location-model structure with sparsity and kernel smoothing, rather than fitting an unrestricted or purely low-dimensional discriminant rule.
>
> Regarding KNN, we agree that it is a natural nonparametric benchmark. In fact, KNN can be viewed as a nearest-neighbor/local-smoothing method, and is therefore conceptually related to kernel smoothing. However, the two methods use the local information differently. A standard KNN classifier mainly relies on majority voting among the nearest observations and does not use the conditional Gaussian discriminant structure under the location model. Moreover, nearest-neighbor methods can be sensitive to the choice of distance metric in high-dimensional spaces, where the notion of proximity may become less informative (Beyer et al. (1999); Aggarwal et al. (2001)). By contrast, SLM uses kernel smoothing only to estimate the location-specific quantities $\\mu_1(u)$, $\\mu_2(u)$, and $\\Sigma(u)$, and then constructs the discriminant direction
>
> $$
> \\beta(u)=\\Sigma^{-1}(u)\\{\\mu_1(u)-\\mu_2(u)\\}.
> $$
>
> Thus, when the location model is a reasonable approximation and the interaction between the categorical profile and the continuous variables is important, SLM is a more structured alternative to a direct KNN classifier.

---

> ### Author Response · Authors · 2026-06-12
> **Response to reviewer hrWB (part 2)**
>
> > Theoretical results should be further clarified. The theoretical results need further clarification. When kernel smoothing is involved, the smoothness conditions (C3) are not presented in the standard way. Could the authors state the Hölder or Lipschitz smoothness conditions on the true functions explicitly?
>
> **Reply:** We thank the reviewer for pointing this out. We agree that the role of Condition (C3) should be explained more clearly. The main reason why (C3) is not stated in the standard Euclidean Hölder/Lipschitz form is that the smoothing variable in our problem is the high-dimensional discrete vector $U\\in\\{0,1\\}^d$. In this setting, there is no fixed-dimensional Euclidean neighborhood or Lebesgue density with respect to which the usual kernel-smoothing smoothness condition can be directly stated. As discussed after Condition (C3), the point mass probability at a fixed location $u$ can be exponentially small in $d$, and the relevant object is the normalized kernel-weighted local bias.
>
> More specifically, (C3) controls the bias term
> $$
> \\kappa_u(t) = \\frac{ E[m(U)-m(u)]\\exp\\{-(dt)^{-1}N_u\\} }{ E\\exp\\{-(dt)^{-1}N_u\\} }, \\qquad \\kappa(t)=\\sup_{u\\in\\{0,1\\}^d}\\kappa_u(t),
> $$
>  where $N_u=|U-u|_1$. This is the high-dimensional discrete analogue of the usual smoothness condition used to control the bias of a kernel estimator. The same quantity appears in Theorems 3.1 and 3.2 through the term $\\kappa(h)$.
> We also note that there are three examples after Condition (C3) to make this condition more concrete. In particular, Example 2, titled "Lipschitz with exponential order", and Example 3, titled "Centered Lipschitz with exponential order", are intended to give Lipschitz-type interpretations of (C3) for high-dimensional discrete variables.
>
> However, we agree that this connection was not made sufficiently explicit. In particular, we added the following explanation before the examples:
> > "Condition (C3) plays the same role as a smoothness condition in classical kernel smoothing: it controls the local bias of the kernel estimator. Since the smoothing variable $U$ is a high-dimensional binary vector, a standard fixed-dimensional Hölder or Lipschitz condition is not directly applicable. Instead, (C3) is formulated through the normalized kernel-weighted difference between $m(U)$ and $m(u)$ over the Hamming cube. The following examples illustrate how (C3) can be verified. In particular, Examples 2 and 3 can be viewed as Lipschitz-type sufficient conditions adapted to high-dimensional discrete variables."

---

> ### Author Response · Authors · 2026-06-12
> **Response to reviewer hrWB (part 3)**
>
> > Numerical results. The interpretation of the numerical results should be more careful and reliable. Section 4.3 states: "in Table 1, from which we observe that our proposed SLM classifier outperforms other classifiers." However, under the first three configurations of Model 1 and the last three configurations of Model 2, the DSDA method outperforms the proposed SLM. Moreover, the PLG method also yields better performance in some settings. There is no clear pattern indicating when the proposed method is superior. The authors may increase the number of Monte Carlo repetitions and provide a more detailed discussion of the empirical advantages of their method.
>
> **Reply:**
>
> We thank the reviewer for this careful comment. We agree that the original interpretation of the numerical results was too strong. In the revised manuscript, we no longer state that the proposed SLM uniformly outperforms all competing methods. We have revised the wording in Section 4.3 and now present a more careful comparison.
>
> We have also updated the simulation study in Table 1. First, the number of replications has been increased from 100 to 200. Second, the best empirical result in each configuration is highlighted in boldface, so that the comparison across methods is more transparent. Third, we have added additional benchmark methods, including factor analysis of mixed data (FAMD), gradient boosting (GB), and mixed-type naive Bayes (MNB).
>
> The revised Table 1 gives a clearer picture. SLM often attains the smallest empirical misclassification rate and performs close to the Bayes risk in many configurations. However, it is not uniformly the best method. For example, RF gives the smallest error in several configurations, and GB gives the smallest error in one configuration of Model 3. This confirms the reviewer's point that the numerical results should not be interpreted as uniform dominance of SLM over all alternatives. Instead, the results show that SLM is a competitive classifier for high-dimensional mixed variables under the location-model simulation settings considered in the paper.
>
> We also revised the discussion of the real-data study in Table 2. The results again show that SLM and its variants are competitive: SLM or its two variants achieve the best error in several data sets, while RF and GB are better in some others. We therefore describe the empirical advantage of SLM more cautiously as its stable and competitive performance across both simulated and real mixed-variable data, rather than as universal superiority.
>
> Specifically, we replaced the original sentence
>
> > "in Table 1, from which we observe that our proposed SLM classifier outperforms other classifiers."
>
> by the following revised interpretation:
>
> > "Table 1 reports the mean and standard deviation of the misclassification rates over 200 replications. The best empirical result in each configuration is highlighted in boldface. Overall, SLM is competitive across the four simulation models and often attains the smallest empirical misclassification rate, with performance generally close to the Bayes risk. However, SLM is not uniformly the best method in every configuration. For example, RF and GB achieve smaller errors in some settings. The results should therefore be interpreted as evidence that SLM is a competitive classifier for high-dimensional mixed variables under the location-model settings considered here, rather than as evidence of uniform dominance over all competing methods."

---

> ### Author Response · Authors · 2026-06-12
> **Response to reviewer hrWB (part 4)**
>
> > 1.What is the difference between equation (3) and equation (7)?
>
> **Reply:** We thank the reviewer for pointing out this possible ambiguity. Equation (3) is the Bayes classifier under the semiparametric location model, written in terms of the true model quantities
> $$
> \\beta(u)=\\Sigma^{-1}(u)\\{\\mu_1(u)-\\mu_2(u)\\}, \\qquad \\eta(u)=\\log\\{\\pi_1p_1(u)/(\\pi_2p_2(u))\\}.
> $$
>  In contrast, equation (7) introduces a general discriminant rule indexed by arbitrary functions $b(u)$ and $b_0(u)$. It is not a different model assumption; rather, it is a general expression used to study the risk-minimization problem. Proposition 2.1 then connects these two objects: it shows that the true direction $\\beta(u)$ appearing in the Bayes classifier (3) can be characterized through the risk minimization of the zero-intercept subclass $D(b(u),0)$ of the general rule in (7), while the true intercept is $\\eta(u)$. We have added the following clarification immediately after equation (7):  "Here (7) is a general discriminant rule with arbitrary functions $b(u)$ and $b_0(u)$. It should be distinguished from (3), which is the Bayes classifier under the location model and is written in terms of the true quantities $\\beta(u)$ and $\\eta(u)$.  "
>
> > 2.Highlight the best results in Table 1.
>
> **Reply:** We have revised Table 1 to highlight the best result in each setting in boldface. The caption of Table 1 has been revised accordingly.
>
> The revised caption reads as: "The best result in each setting is highlighted in boldface."
>
>
> > 3.Please provide a pseudo algorithm summary of the proposed method.
>
> **Reply:** We have added Algorithm 1 at the end of Section 2 to summarize the proposed method. The algorithm includes kernel estimation of $\\mu_1(u)$, $\\mu_2(u)$ and $\\Sigma(u)$, sparse estimation of $\\beta(u)$, sparse logistic estimation of $\\eta(u)$, tuning selection, and the final classification rule.
>
> The added algorithm is as follows.
>
> > Algorithm 1: Semiparametric location classifier.
> >
> > Input: Training data $\\{(U_i,X_i),L_i=1\\}_{i=1}^{n_1}$ and $\\{(V_j,Y_j),L_j=2\\}_{j=1}^{n_2}$; tuning grids for the smoothing parameter and penalties.
> >
> > Step 1. For each target location $u$, compute kernel weights on the Hamming cube and estimate $\\mu_1(u)$, $\\mu_2(u)$, and $\\Sigma(u)$ by the normalized kernel estimators.
> >
> > Step 2. Estimate the location-dependent discriminant direction by
> >
> > $$
> > \\widehat\\beta(u)=\\arg\\min_{b\\in\\mathbb R^p}
> > \\left\\{ b^T\\widehat\\Sigma(u)b-2b^T(\\widehat\\mu_1(u)-\\widehat\\mu_2(u))+\\lambda_\\beta |b|_1\\right\\}.
> > $$
> >
> >
> > Step 3. Estimate the intercept $\\eta(u)$ by fitting the sparse logistic model
> >
> > $$
> > \\widehat\\eta(u)=\\widehat A_0+\\widehat A^T u.
> > $$
> >
> >
> > Step 4. Select tuning parameters by cross-validation using the misclassification criterion described in Section 4.1.
> >
> > Step 5. For a new observation $(Z,u)$, classify it to class 1 if
> >
> > $$
> > \\widehat\\beta(u)^T\\left\\{Z-\\frac{\\widehat\\mu_1(u)+\\widehat\\mu_2(u)}{2}\\right\\}+\\widehat\\eta(u)>0,
> > $$
> >
> > and to class 2 otherwise.

---

### Decision · Action_Editor_yJ2G · 2026-07-06

**Recommendation:** Accept as is

**Comment:**

Both reviewers recommended acceptance after revision. Their concerns were fully addressed: the authors clarified when the location-model structure is preferable to logistic regression and KNN, gave a Lipschitz-type interpretation of condition (C3) adapted to the high-dimensional discrete setting, increased the Monte Carlo replications from 100 to 200, added FAMD, GB, and MNB as competitors, and revised the overstated uniform-superiority claim in Section 4.3 to a more calibrated statement. They also added Algorithm 1, discussed the common-covariance restriction as inherited from LDA (with QDA-type extensions left to future work), and addressed multi-category and ordinal encoding. The theoretical contribution is solid and the empirical claims are now appropriately calibrated. I recommend acceptance.

**Audience:**

yes

**Claims And Evidence:**

yes